# Structural basis for the interaction between the bacterial cell division proteins FtsZ and ZapA

Junso Fujita [1,2,3,8], Kazuki Kasai [1,2,8], Kota Hibino[4], Gota Kagoshima[5], Natsuki Kamimura[4], Shungo Tobita [4], Yuki Kato[4], Ryo Uehara[4], Keiichi Namba [1,2], Takayuki Uchihashi [5,6,7] & Hiroyoshi Matsumura [4]

Cell division in most bacteria is regulated by the tubulin homolog FtsZ as well as ZapA, a FtsZ-associated protein. However, how FtsZ and ZapA function coordinately has remained elusive. Here we report the cryo-electron microscopy structure of the ZapA-FtsZ complex at 2.73 Å resolution. The complex forms an asymmetric ladder-like structure, in which the double antiparallel FtsZ protofilament on one side and a single protofilament on the other side are tethered by ZapA tetramers. In the complex, the extensive interactions of FtsZ with ZapA cause a structural change of the FtsZ protofilament, and the formation of the double FtsZ protofilament increases electrostatic repulsion. High-speed atomic force microscopy analysis revealed cooperative interactions of ZapA with FtsZ at a molecular level. Our findings not only provide a structural basis for the interaction between FtsZ and ZapA but also shed light on how ZapA binds to FtsZ protofilaments without disturbing FtsZ dynamics to promote cell division.

Cell division in nearly all bacteria is initiated by the polymerization of the tubulin homolog FtsZ at midcell[1–3]. In the presence of GTP, FtsZ polymerizes into protofilaments, which further associate into a ring-like structure (the Z-ring). The Z-ring has two important functions: recruiting the cell division proteins as a scaffold, and treadmilling of FtsZ protofilaments dependent on its GTPase activity. Treadmilling is a motion in which the FtsZ molecule binds to the protofilament's end (plus-end) and depolymerizes at the other end (minus-end). FtsZ treadmilling is critical to promote condensation of diffuse FtsZ protofilaments into a coherent Z-ring, and also for stimulating septal cell wall synthesis, which is essential for cell constriction[4,5].

FtsZ-associated proteins (Zaps) such as ZapA, ZapB, ZapC, and ZapD facilitate Z-ring assembly[6–9]. *E. coli* mutants lacking one of the Zap genes typically exhibit elongated cells, and *E. coli* mutants lacking multiple Zap genes show mislocalized and distorted Z-rings, resulting in a more severe phenotype and/or death. The best-characterized Zap is ZapA, which is conserved widely in gram-negative and gram-positive bacterial species[6,10,11]. ZapA is usually expressed at a very high level in bacterial cells; for example, the cellular ZapA concentration in *E. coli* is estimated to be 5.4 µM, approximately equal to that of FtsZ[11]. FtsZ can assemble into helical or straight protofilaments in vivo[12–14] and in vitro[15]. ZapA binds FtsZ straight protofilaments preferentially to crosslink FtsZ protofilaments[16]; thereby, ZapA bundles more aligned and straight filaments constituting a coherent Z-ring[11,17]. A recent in vitro reconstituted FtsZ and ZapA study has revealed that ZapA binds to FtsZ protofilaments in a highly cooperative manner, and has no effect on treadmilling velocity[17]. Such interaction between FtsZ and ZapA is suggested to be important in maintaining a coherent and dynamic Z-ring. Thus, along with FtsZ, the importance of the ZapA function in bacterial cell division has gained attention.

[1]Graduate School of Frontier Biosciences, University of Osaka, Osaka, Japan. [2]JEOL YOKOGUSHI Research Alliance Laboratories, University of Osaka, Osaka, Japan. [3]Graduate School of Pharmaceutical Sciences, University of Osaka, Osaka, Japan. [4]Department of Biotechnology, College of Life Sciences, Ritsumeikan University, Kusatsu, Japan. [5]Department of Physics, Nagoya University, Nagoya, Japan. [6]Exploratory Research Center on Life and Living Systems (ExCELLS), National Institutes of Natural Sciences, Okazaki, Japan. [7]Institute for Glyco-core Research (iGCORE), Nagoya University, Nagoya, Japan. [8]These authors contributed equally: Junso Fujita, Kazuki Kasai. ✉e-mail: uchihast@d.phys.nagoya-u.ac.jp; h-matsu@fc.ritsumei.ac.jp

Structures of FtsZ and ZapA from several bacterial species have been characterized independently. Recently, we have reported the high-resolution cryo-electron microscopy (cryo-EM) structure of *K. pneumoniae* FtsZ protofilaments, demonstrating the structural details of straight and helical protofilaments[15]. Crystallographic analyses of ZapA from *P. aeruginosa* and *E. coli* have been performed, showing that ZapA forms a dumbbell-like tetramer with domains at both ends that bind to FtsZ[10,18]. The ladder-like structure of the *E. coli* ZapA-FtsZ complex has been previously observed by negative staining electron microscopy[11,19]. Despite extensive studies, our understanding of ZapA-FtsZ coordination has been hampered by a lack of a high-resolution structure of the ZapA-FtsZ complex.

In this study, we report the high-resolution cryo-EM structure of the *K. pneumoniae* ZapA-FtsZ complex. The cryo-EM structure of the ZapA-FtsZ complex reveals an asymmetric ladder-like architecture, in which the double antiparallel FtsZ protofilament on one side and a single protofilament on the other side are tethered by ZapA tetramers. The extensive interactions of FtsZ with ZapA cause a structural change in the FtsZ protofilament, shedding light on how ZapA cooperatively binds to FtsZ protofilaments. The electrostatic repulsion is found in the inter-filament interactions within the FtsZ double protofilaments, possibly explaining why ZapA binding does not affect FtsZ treadmilling velocity. Structure-based mutagenesis revealed that interactions via the FtsZ N-terminal tail are essential for forming the ZapA-FtsZ complex. Cryo-EM re-analysis focusing on ZapA, including the ZapA dimeric mutant (I83E)-FtsZ complex, revealed multiple distinct interaction patterns between ZapA and single FtsZ protofilaments, suggesting that the ZapA-FtsZ interaction is highly dynamic and fluctuating. Despite this variability, the FtsZ N-terminal tail interactions were consistently preserved. High-speed atomic force microscopy (HS-AFM) further revealed positive cooperativity of the interaction of ZapA and FtsZ, and stronger affinity between ZapA and FtsZ, especially when adjacent ZapA molecules were present on double protofilaments. HS-AFM also captured how the ZapA-FtsZ complex forms. Taken together, our data provide a structural basis for the interaction between FtsZ and ZapA and lead to a mechanistic model for regulating bacterial cell division.

## Results

### The ZapA tetramer binds to antiparallel double protofilaments of FtsZ

We first observed the *K. pneumoniae* ZapA-FtsZ complex by negative staining EM and found many twisted ladder-like structures (Supplementary Fig. 1a), as observed previously[11,19]. To determine the high-resolution structure, we performed cryo-EM data collection of the ZapA-FtsZ complex. Many paired filaments with aligned high-contrast dots were observed (Fig. 1a), and 2d class averages indicated the ladder-like structure (Fig. 1b). During picking by filament tracer, it quite often happened that one side of the filament in the ladder was centered instead of the middle of the ladder with a lower signal. Therefore, the reconstructed map showed clear features on one side of the ladder, but blurred features on the opposite side (Supplementary Fig. 1b). We therefore re-centered the initial 3D map and re-extracted particles for another reconstruction to observe the filaments on both sides of the ladder. The reconstructed map showed an asymmetric ladder structure; the FtsZ formed a double protofilament on one side, and a single protofilament on the other side, and these were tethered by ZapA tetramers (Supplementary Fig. 2a, b). The axes along the two FtsZ protofilaments were not completely parallel. Even after many trials, we failed to improve the resolution of the FtsZ single protofilament region, and therefore, the polarity and orientation of the FtsZ single protofilament could not be determined. Hereafter, we focused on the FtsZ double protofilament bound to ZapA tetramer. We subtracted the single protofilament region and performed symmetry expansion with C2 symmetry and 3D classification to align "upside-down" double

protofilaments. After removing the duplicated particles, we performed helix refine with D1 symmetry, namely with a single C2 axis perpendicular to the helical axis but no rotational symmetry along the helical axis[20], and reached an overall map resolution of 2.73 Å (Supplementary Fig. 3a–c) with an optimized helical rise of 44.58 Å and a helical twist of −3.11° (Supplementary Table 1). The well-resolved FtsZ double protofilament structure enabled us to confirm that all FtsZ monomers are in the T conformation with GMPCPP bound and that the two FtsZ protofilaments are antiparallel (Fig. 1c, d). Secondary structural elements are mostly conserved in ZapA-free and ZapA-bound FtsZ (Supplementary Fig. 4a). One dimer head of ZapA was located among four FtsZ molecules. The dimer head of ZapA on the other side could not be modeled.

### Two ZapA molecules interact with four FtsZ molecules through the N-terminal tail of FtsZ

In the ZapA-FtsZ interface, the ZapA dimer head (chains G, H) interacted with two pairs of FtsZ dimers from two antiparallel protofilaments (chains A, B, and E, F) (Fig. 1d). As the ZapA-FtsZ complex has a C2 axis perpendicular to the protofilament, we focused on the interactions around one pair of FtsZ dimers (chains A, B). The N-terminal region of FtsZ (residues 1–10) was sandwiched between two ZapA molecules (Fig. 2a). The region from Leu41 to Tyr63 in one ZapA (chain G) mainly interacted with the N-terminal tail from Met1 to Thr8 in FtsZ (chain A) (Supplementary Fig. 5a). Thr45, Arg46, Val47, Thr48, and Asn60 in ZapA (chain G) formed an elaborate network of hydrogen-bonding interactions with Phe2, Met5, Glu6, and Leu7 in FtsZ (chain A). The region from Arg16 to Gln23 in the other ZapA (chain H) primarily interacted with the same N-terminal tail of FtsZ (chain A). Hydrogen bonds were observed between Gln23 in ZapA (chain H) and Met1 in FtsZ (chain A), and between Asn18 in ZapA (chain H) and Glu3 of FtsZ (chain A). The region from Asp7 to Arg16 in the same ZapA (chain H) interacted with the region from Arg33 to Gly55 in the other FtsZ (chain B) (Supplementary Fig. 5b). Gly12, Ser14, and Arg16 in ZapA (chain H) formed hydrogen-bonding interactions with Arg33, Thr52, Ala53, Val54, and Gly55 in FtsZ (chain B). The side chain of Asn49 in ZapA (chain G) made a hydrogen bond with the carbonyl oxygen of Lys51 in FtsZ (chain B).

Protein-protein interactions were analyzed using the PDBe PISA server[21] (Supplementary Table 2). The interface area of the ZapA dimer head (chains G, H) with two pairs of FtsZ dimers is extensive (2105.4 Å²) with a $\Delta G$ of dissociation of −17.0 kcal/mol, as compared with the interface area and a $\Delta G$ of dissociation between two FtsZ molecules within a FtsZ protofilament, which are 1390.6 Å² and −16.0 kcal/mol, respectively.

ZapA tetramers were stabilized by the bundling of four C-terminal helices. In this analysis, up to Ile90 was resolved in the ZapA dimer bound to the double protofilaments of FtsZ (Fig. 2b). In the other ZapA dimer, only residues 85–103 could be built, and the dimer head was missing due to weak density (Fig. 1c, d). To acquire the structural information of the missing dimer head, we determined the crystal structure of *K. pneumoniae* ZapA (KpZapA) at 1.8 Å resolution (Fig. 2c, Supplementary Table 3, Supplementary Figs. 4b and 6a) and performed a structural comparison of ZapA within the ZapA-FtsZ complex with the crystal structures of KpZapA, *E. coli* ZapA (EcZapA, PDB code: 4P1M[18]), and *P. aeruginosa* ZapA (PaZapA, PDB code: 1W2E[10]). The cryo-EM structure of KpZapA within the ZapA-FtsZ complex was superimposed well to the crystal structures of KpZapA and EcZapA (r.m.s.d. = 0.851 Å among 128 $C_\alpha$ atom for KpZapA, and 0.911 Å among 127 $C_\alpha$ atom for EcZapA) (Fig. 2c). For the dimer head on the missing side, KpZapA and EcZapA overlapped well, but PaZapA did not. The N-terminal tail of FtsZ in our cryo-EM structure, which is sandwiched by two ZapA molecules (Fig. 2a), overlaps with the C-terminal tail of KpZapA and EcZapA (Supplementary Fig. 6b). This observation indicates that the C-terminal tail of ZapA moves aside to accept the

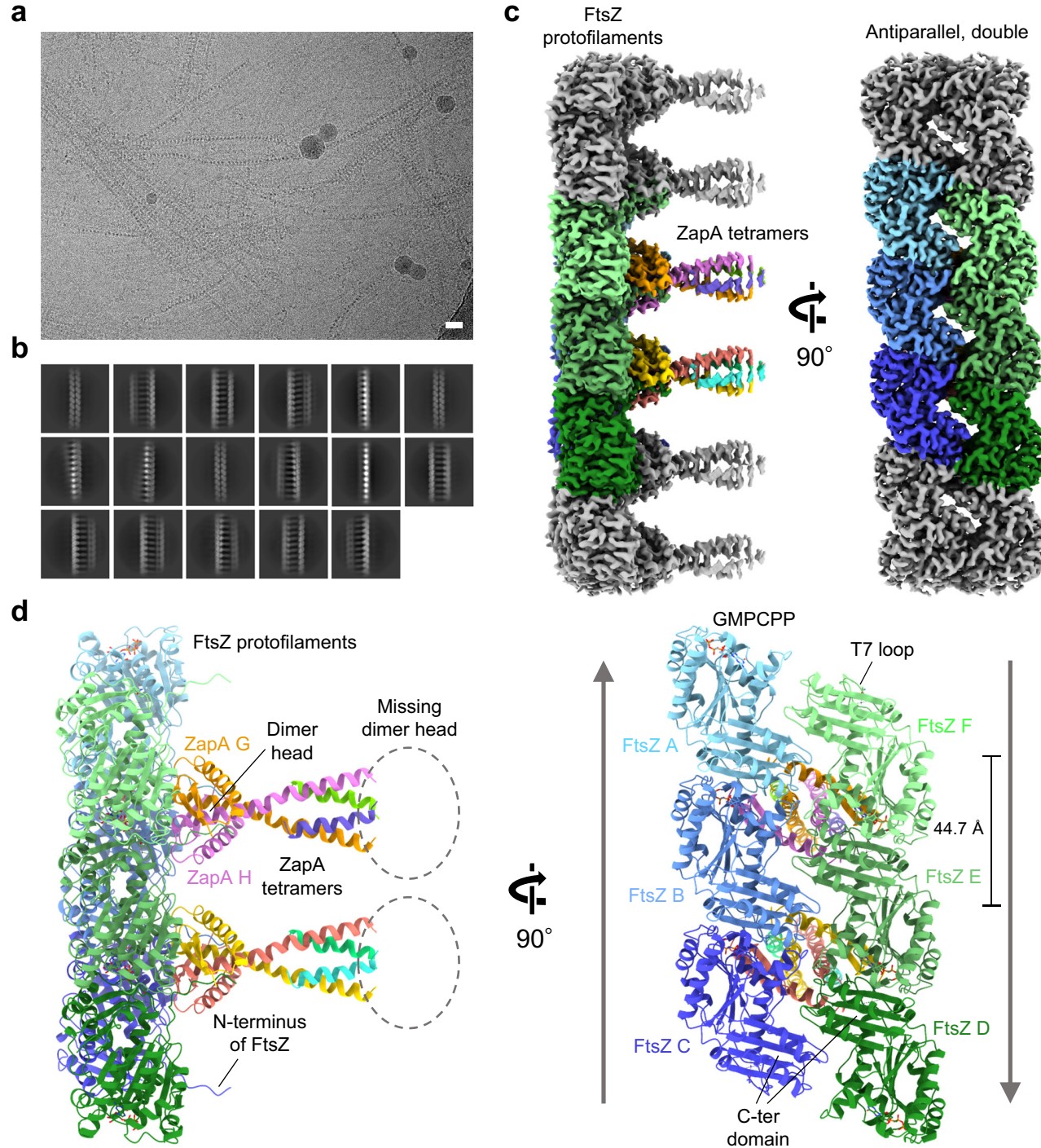

**Fig. 1 | Overall cryo-EM structure of an antiparallel double filament of KpFtsZ in complex with KpZapA tetramer. a** Typical raw micrograph of the ZapA-FtsZ complex supplemented with 1 mM GMPCPP. The scale bar represents 20 nm. Similarly, 6070 micrographs were collected (Supplementary Fig. 1). **b** 2D class averages selected before ab initio reconstruction. **c** Final sharpened map at an overall resolution of 2.73 Å. The colored region corresponds to the region we reconstructed the model, and each color represents different chains. **d** The reconstructed model containing six FtsZ monomers and two ZapA tetramers. The coloring is the same as in (**c**). The directions of two antiparallel FtsZ protofilaments are shown in the right panel.

N-terminal tail of FtsZ when the ZapA-FtsZ complex forms. Furthermore, the α1-α2 loop of ZapA undergoes a conformational change to recognize the N-terminal tail of FtsZ.

Based on the structure, we analyzed the interaction through the N-terminal tail of FtsZ by mutagenesis. We first manually replaced the FtsZ's N-terminal residue with alanine and estimated its contribution by calculating the ΔG of dissociation using the PDBe PISA server[21] (Supplementary Table 4). Compared with the wild-type FtsZ, we

identified Phe2 as the most significant contributor among the N-terminal residues and generated the FtsZ-F2A mutant (F2A). Sedimentation assays confirmed its inability to interact with ZapA (Supplementary Fig. 7a), and HS-AFM observations also revealed that F2A lost its ability to bind ZapA (Supplementary Fig. 7b, c and Supplementary Movies 1, 2). Phe2 is highly conserved in Gram-negative bacterial FtsZ, while in Gram-positive bacteria, a phenylalanine is conserved at the fourth position from the N-terminus (Supplementary

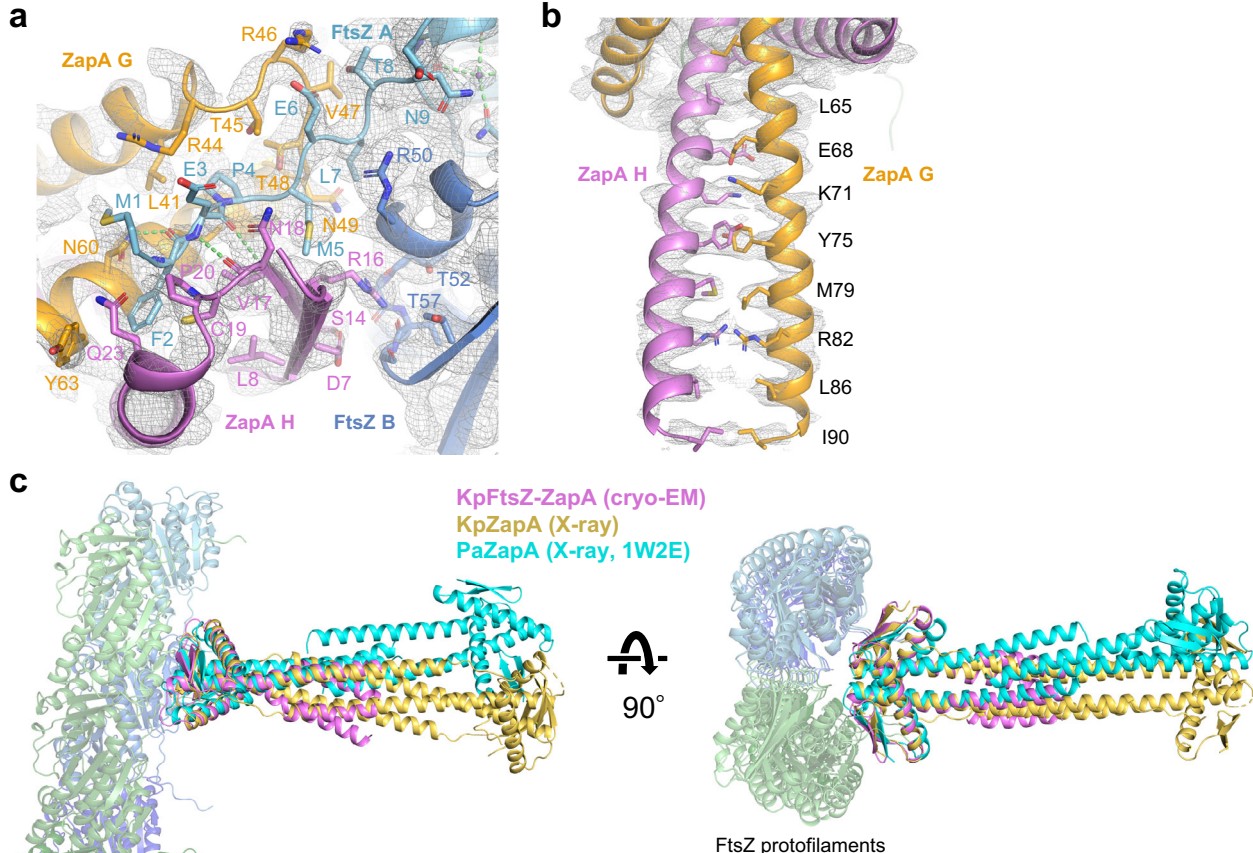

**Fig. 2 | Interactions between FtsZ and ZapA. a** Interface between FtsZ and ZapA. N-terminus of FtsZ is located between two ZapA monomers. Final sharpened map is represented as a gray mesh. Hydrogen bonds are shown as green dashes. **b** Bundled helices within one ZapA dimer. Another ZapA dimer is not shown for clarity. **c** Superimposition of cryo-EM structure of the ZapA-FtsZ complex (violet) and crystal structures of KpZapA (yellow, this study) and PaZapA (cyan, PDB code: 1W2E). The structures are superimposed based on ZapA dimers.

Fig. 4a), implying a similar function given the flexibility of the N-terminal tail. These findings propose a mechanism where the FtsZ N-terminal tail captures ZapA via Phe2, like a hook.

**Variations in interactions between ZapA and FtsZ**

To obtain structural information on the single FtsZ protofilament side within the ZapA-FtsZ complex, we re-analyzed the same cryo-EM data of the ZapA-FtsZ complex focused on ZapA (Supplementary Fig. 8). As described, the structures of half of a ZapA molecule and a single protofilament could not be constructed in the above-described high-resolution cryo-EM analysis, because the density map was rather blurred on the single protofilament side. Re-analysis focused on ZapA allowed us to visualize the whole ZapA molecule and single FtsZ protofilament structures at a level where the secondary structures can be recognized by 2D class images (Supplementary Fig. 9). The variations in the angle are found between the FtsZ double protofilament axis and the long axis of the coiled-coil domain of ZapA tetramer (red lines in Supplementary Fig. 9a), Notably, in the rightmost image of Supplementary Fig. 9a, the ZapA head domain is almost detached from the FtsZ double protofilaments, suggesting that the extended N-terminal tail of FtsZ barely maintains its binding and this interaction is flexible. Such flexible interaction explains why a high-resolution 3D reconstruction of this portion was unsuccessful. But, by focusing on ZapA, recognizable secondary structures on the single protofilament side allowed us to determine the azimuthal orientations of the protofilament and ZapA (Supplementary Fig. 9b). The orientation of the ZapA dimer head relative to the single protofilament is roughly perpendicular to that bound to the double protofilaments, emphasizing that the

ZapA-FtsZ complex represents an asymmetric structure. On this side, the ZapA dimer heads are positioned near the FtsZ N-terminus, suggesting that the N-terminal tail commonly participates in ZapA binding.

To further investigate how the ZapA dimer interacts with the FtsZ protofilament, we introduced the I83E mutation into ZapA, and analyzed the ZapA(I83E)-FtsZ complex using cryo-EM (Supplementary Fig. 10). The ZapA(I83E), which has a mutation in the coiled-coil region, forms a dimer with a single dimer head[22] (Supplementary Fig. 11a). The 2D class images showed that ZapA(I83E) binds only sparsely to the single FtsZ protofilament (Supplementary Fig. 11b–e, most left), suggesting that ZapA tetramerization is necessary for stable binding. The FtsZ secondary structure matched that in Supplementary Fig. 9b, with cloud-like densities appearing where ZapA would be located (Supplementary Fig. 11b). In the average images (Supplementary Fig. 11c, d), these densities were positioned near the FtsZ N-terminal tail. In Supplementary Fig. 11e, the density of an FtsZ subunit is enhanced, where the ZapA density overlaps the FtsZ density, likely due to ZapA binding to its extended N-terminal tail. These results indicate that ZapA(I83E) binds to the FtsZ face containing the N-terminal tail, similarly to ZapA binding to single and double protofilaments. This supports the findings of the re-analysis focused on ZapA.

**Intra- and inter-FtsZ protofilament interactions**

We found GMPCPP in the binding pocket between two FtsZ molecules in the protofilament (Fig. 3a). The γ-phosphate was located near the T3 loop, and similar conformations of GMPCPP have been observed in the previous crystal structures of *S. aureus*

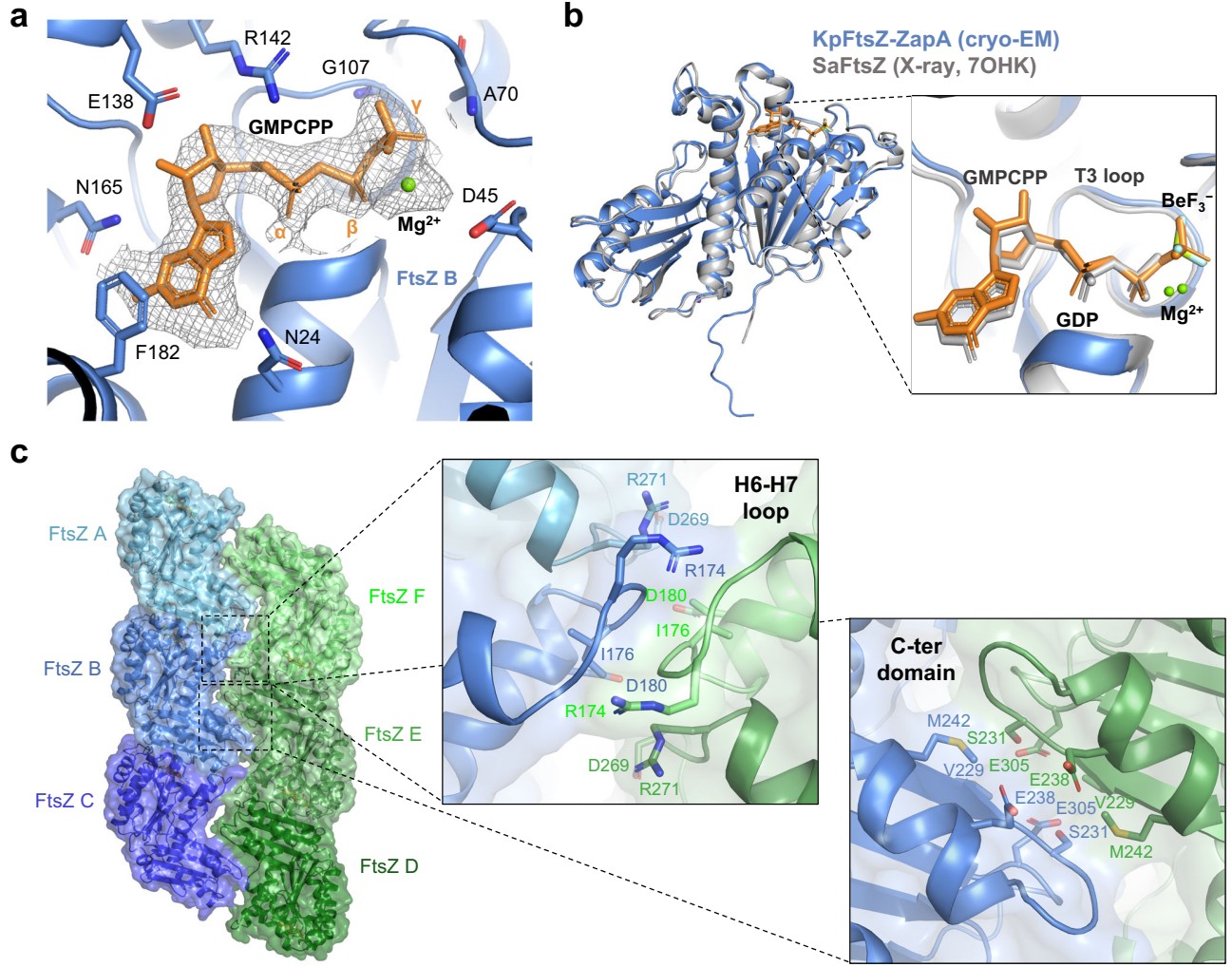

**Fig. 3 | Interactions in FtsZ protofilaments. a** Close-up view around GMPCPP bound to FtsZ. The final sharpened map is shown as a gray mesh and drawn only around GMPCPP. **b** Superimposition of KpFtsZ within the KpZapA-FtsZ complex with *S. aureus* FtsZ complexed with GDP, BeF$_3^-$, and Mg$^{2+}$ (PDB code: 7OHK). **c** Interfaces between two antiparallel FtsZ protofilaments. Surface of FtsZ protofilaments is also shown. The right panels are close-up views around the H6-H7 loop and C-terminal domain. ZapA is not shown for clarity.

FtsZ complexed with GDP, BeF$_3^-$, and Mg$^{2+}$ (PDB code: 7OHK[23]) (Fig. 3b), suggesting mimicking the transition state of GTP hydrolysis. The H6-H7 loop and the C-terminal domain mainly contributed to the interaction between the two antiparallel protofilaments of FtsZ (Fig. 3c).

Inter-protofilament interactions within the double FtsZ filaments appear to be weak. According to the PISA calculation, chain B and chains F and E form an interaction area of 523.4 Å$^2$ with a $\Delta G$ of dissociation of −3.6 kcal/mol (Supplementary Table 2). This feature is rationalized by the electrostatic repulsions among Glu238 and Glu305 found in the inter-protofilament interactions (Fig. 3c).

The entire structure of the FtsZ protofilament with ZapA was roughly similar to that without ZapA (r.m.s.d. = 1.53 Å among 918 C$_\alpha$ atoms for three KpFtsZ molecules within a protofilament), but ZapA binding induces structural changes in a FtsZ molecule as well as an adjacent FtsZ molecule. When comparing the structures of FtsZ within the ZapA-FtsZ complex and that of FtsZ straight protofilaments alone (PDB code: 8IBN[15]), approximately 2–3 Å shifts of the β2-α2 loop and the α1-β2 loop are observed (indicated by red and green arrows in Fig. 4). The β-sheet comprising β-strands 1–5 slightly shifts upon ZapA binding (indicated by a blue arrow in Fig. 4). The side chains of Asp209 and Phe210 in FtsZ also rearrange.

## HS-AFM analysis of the ZapA-FtsZ complex

To investigate the single-molecule dynamics of the ZapA-FtsZ complex, we performed HS-AFM analysis. For HS-AFM observations, two approaches were used: either pre-mixing FtsZ and ZapA in a tube before adsorption onto a mica substrate (referred to as "pre-mixing method"), or first adsorbing only FtsZ onto the mica substrate, followed by the addition of ZapA to the observation solution (referred to as "ZapA addition method"). We first observed the ZapA-FtsZ complex using the pre-mixing method, and it revealed ladder-like structures in which ZapA molecules tether to the FtsZ protofilaments (Fig. 5a–c). Judging from the measured heights from the cross-sectional profile and their height histograms (Fig. 5d and Supplementary Fig. 12), ZapA appeared to bridge two single FtsZ protofilaments on the mica substrate to form ladder-like structures. As far as we observed by HS-AFM using the pre-mixing method, the structures of ZapA bridging double FtsZ protofilaments and single protofilaments, as resolved by cryo-EM, were not observed (discussed in detail later).

Next, we observed the ZapA-FtsZ complex using the ZapA addition method. When only FtsZ was absorbed into the mica substrate (Fig. 5e), the FtsZ protofilaments fluctuated somewhat on the mica substrate (Supplementary Movie 3). After the addition of ZapA, bright spots were clearly observed on the FtsZ protofilaments (Fig. 5f). A

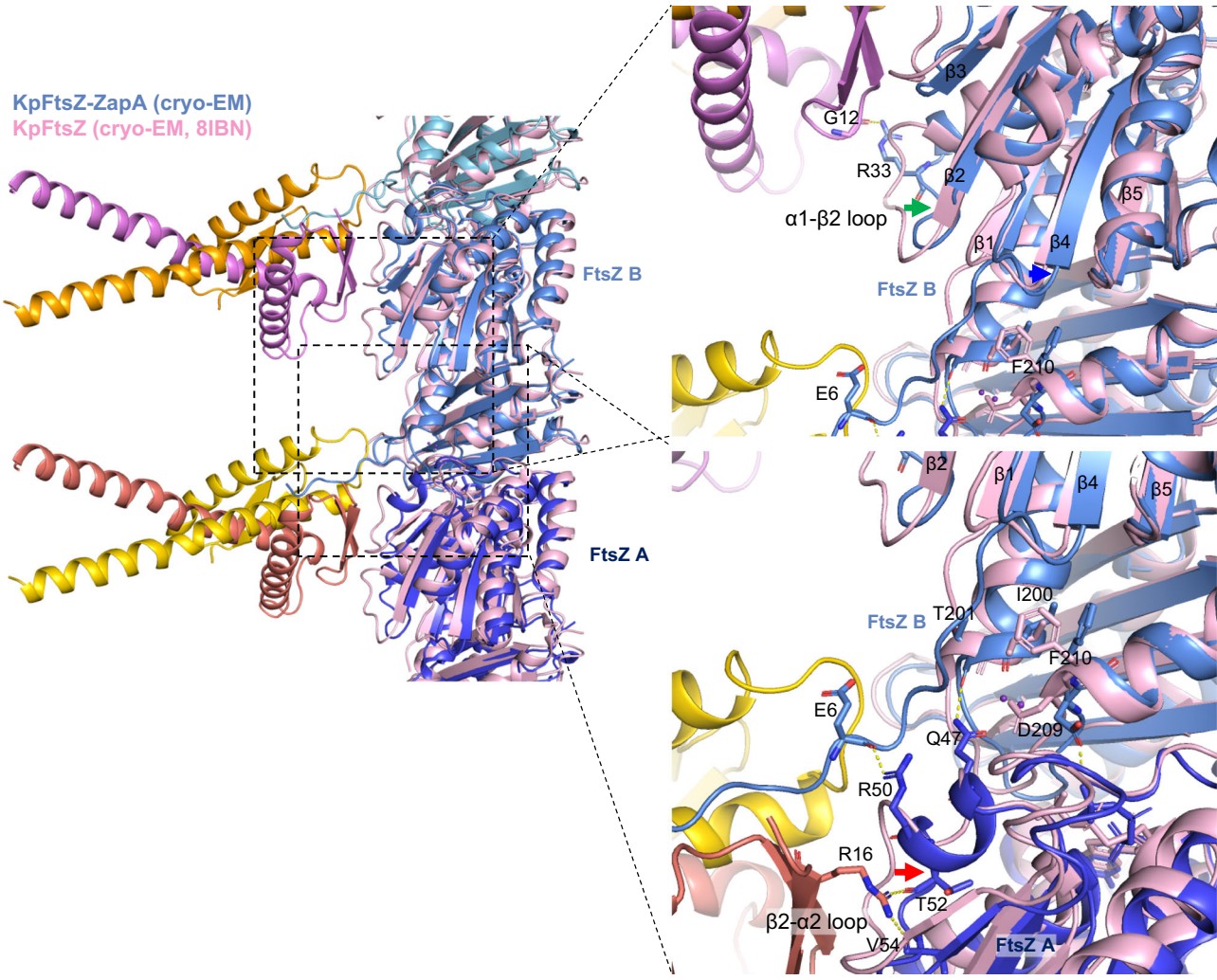

**Fig. 4 | Superimposition of the ZapA-FtsZ complex and the KpFtsZ straight protofilaments alone.** The coloring of the KpZapA-FtsZ complex is the same as in Fig. 1d, and the KpFtsZ straight protofilament alone is colored in pink. Several key residues at the KpZapA-FtsZ interface are shown as sticks and labeled. The red, blue, and green arrows indicate the structural shifts or changes of β2-α2 loop, β-sheet comprising of β1–5, and α1-β2 loop, respectively.

magnified view shows that ZapA binds to the hollow sites between two FtsZ protofilaments as well as on top of a single FtsZ protofilament (Fig. 5g). This is considered to correspond to the binding of ZapA to FtsZ double protofilaments and single protofilaments revealed by cryo-EM. This suggests that the adsorption surface of FtsZ to the mica substrate is opposite to the ZapA binding surface.

The binding of ZapA to the FtsZ protofilaments is not static but dynamic, with repeated binding and dissociation (Fig. 5h and Supplementary Movies 4, 5). As seen in Fig. 5h, there are areas where ZapA bright spots are contiguous along a protofilament, especially when bound to double protofilaments. Therefore, we analyzed the binding residence time of a single bright spot (corresponding to a ZapA tetramer) on FtsZ protofilaments. The residence time of isolated ZapA molecules was 0.34 s on single protofilaments and 0.66 s on double protofilaments (1.9× longer) (Fig. 5i). When adjacent ZapA molecules were present on double protofilaments, it increased to 3.8 s (11× longer), indicating a stronger affinity and positive cooperativity, especially in the presence of neighboring ZapA molecules on double protofilaments.

We examined by diluting the premix and observing it on a mica substrate (Supplementary Fig. 13). When examining samples up to 100× dilution, we observed ladder-like ZapA-FtsZ structures in areas with available space at the 50× dilution. On the other hand, when

diluted up to 100×, the FtsZ filaments were hardly adsorbed onto the substrate, and the ladder-like structures could no longer be observed. The growth process of a ladder-like structure on the mica substrate was also observed (Fig. 5j and Supplementary Movie 6). In this process, a short ladder is formed first, and one of the FtsZ protofilaments extends from it, followed by ZapA binding and extension of the other FtsZ protofilament.

## Discussion

Because both FtsZ and ZapA are widely conserved among bacteria and an interplay between FtsZ and ZapA is pivotal for bacterial cell division, the structural information of the ZapA-FtsZ complex is essential to understand the mechanism of bacterial cell division. The cellular ZapA concentration in *E. coli* is approximately 5.4 μM, roughly equal to that of FtsZ[11]. The number of FtsZ molecules in *E. coli* is estimated to be between 3200[24] and 15,000[25]. Consequently, during the initial stage of cytokinesis, FtsZ self-polymerizes and associates into condensed protofilament bundles to form the Z-ring, and ZapA molecules bind to any part of the Z-ring to stabilize it. A FtsZ protofilament has polarity and lacks symmetry, while ZapA is in an equilibrium between homo-dimer and homotetramer, which are related by C2 and D2 symmetry, respectively. How do the symmetric ZapA molecules bundle the asymmetric FtsZ protofilaments at multipoint interactions? We have

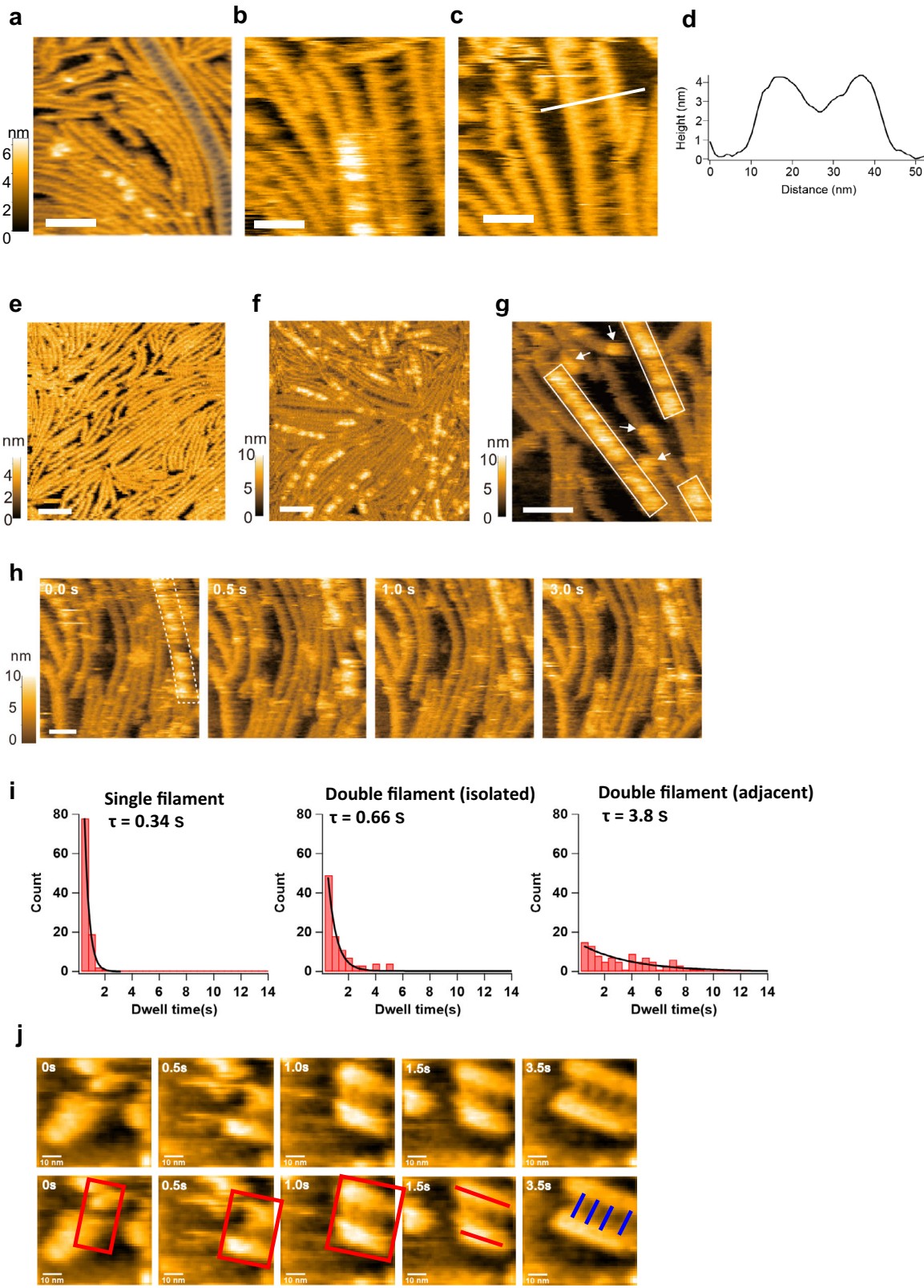

determined the cryo-EM structure of the ZapA-FtsZ complex at high resolution to answer this question. The analysis showed that the double antiparallel FtsZ protofilament on one side and a single protofilament on the other are bridged by ZapA tetramers. Further re-analysis of the same cryo-EM data focusing on ZapA and cryo-EM analysis of the ZapA dimeric mutant (I83E)-FtsZ complex revealed multiple distinct interaction patterns between ZapA and single FtsZ

protofilaments, suggesting the interaction of ZapA with single FtsZ protofilaments is highly dynamic and fluctuating. On the other hand, we must consider the constraints of observations in HS-AFM, where molecules are imaged on a two-dimensional mica substrate. Among the above-described multiple interaction patterns, only molecules compatible with binding on a mica surface should be observed in HS-AFM, where one is the single protofilament-ZapA-single protofilament

**Fig. 5 | HS-AFM observation of the ZapA-FtsZ complex. a** HS-AFM Image of the ZapA-FtsZ complex. The shaded region shows a ladder-like structure where two single FtsZ filaments are cross-linked by ZapA tetramers (pre-mixing method: 1 mM GTP, 30 μM KpZapA). Scale bar: 50 nm, frame rate: 1 fps. **b, c** Enlarged images of the ladder-like structure. **d** The cross-sectional profile along the white line in the image in (**c**). Scale bar: 25 nm, frame rate: 2 fps. 150 × 150 pixels. **e** HS-AFM image of FtsZ filaments alone. Scale bar: 50 nm, frame rate: 1 frame per second (fps), 150 × 150 pixels. **f, g** HS-AFM images of FtsZ filaments interacting with ZapA, **f** pre-mixing method: 0.3 mM GMPCPP, 4.3 μM KpZapA, **g** ZapA addition method: 1 mM GMPPNP, 0.6 μM KpZapA). Scale bar: 50 nm, frame rates: **f** 1 fps, **g** 2 fps, 150 × 150 pixels. Rectangles encircle regions where ZapA is bound to the FtsZ double filaments, and arrows indicate ZapA bound to a single FtsZ filament. **h** Successive HS-

AFM images showing the dynamic interaction between ZapA and an FtsZ filament (ZapA addition method: 1 mM GMPPNP, 0.6 μM KpZapA). Scale bar: 30 nm, frame rate: 2 fps, 135 × 135 pixels. Regions where ZapA is bound to the double filaments are encircled by a broken line at 0 s. **i** Histograms of the dwell time of ZapA bound to the FtsZ filaments under three conditions: on single filaments (left), as an isolated molecule on double filaments (center), and adjacent to another ZapA molecule on double filaments (right) (*n* = 100 for each). **j** Clipped HS-AFM images showing the process of ZapA-mediated crosslinking of FtsZ single filaments. Zap addition method: 1 mM GMPPNP, 1.3 μM KpZapA. Scale bar: 50 nm, frame rate: 2 fps. 150 × 150 pixels. All HS-AFM images shown here are representative images from more than 5 independent experiments. Source data of the graphs are provided as a Source Data file.

configuration (Fig. 5a–c), and the other is ZapA molecules on double or single protofilaments (Fig. 5f–h).

The D2-symmetric ZapA tetramer maintains the nearly parallel alignment of FtsZ protofilaments through solid interactions with the double protofilament and flexible interactions with the single protofilament. As presented in Supplementary Fig. 9, the dimer head orientation on the single protofilament side differs from that on the double protofilament side. However, because the ZapA dimer heads on the single protofilament side are evenly spaced and aligned, ZapA can bind a protofilament via multipoint interactions with the flexible FtsZ N-terminal tail on the single protofilament side. This interaction explains how the three protofilaments are bundled asymmetrically. We propose that the asymmetric and flexible ZapA-FtsZ complex facilitates curvature and dynamic behavior when tethered to the membrane through linker proteins like FtsA or ZipA. Given that FtsZ protofilaments must be moderately curved in the Z-ring, the double and single protofilaments may be positioned on the outer and inner sides, respectively (Fig. 6a). As FtsA forms antiparallel double filaments in the presence of FtsN[26], the antiparallel double filaments of FtsZ stabilized by ZapA may be tethered to those types of FtsA filaments with 1:1 interaction.

Depending on the GTP/GDP ratio, FtsZ adopts monomer conformations and various assembly forms. The GTP-bound FtsZ adopts a straight protofilament form, while the GDP-bound FtsZ can be monomers or curved protofilaments. During the early stage of cell division, GTP triggers FtsZ molecules to form straight protofilaments, and ZapA crosslinks the FtsZ protofilaments to bundle more aligned protofilaments[11,17], constituting a coherent Z-ring. This maturation of the Z-ring is promoted by ZapA's property of preferentially binding to straight FtsZ protofilaments[16]. We consider that this behavior of ZapA may be facilitated by its interaction through the FtsZ N-terminal tail. The monomeric form of FtsZ has a flexible N-terminal tail. In the helical (curved) protofilaments, the FtsZ N-terminal tail adopts a short helix to stabilize the curved conformation, revealed by our previous cryo-EM structure of the GDP-KpFtsZ complexed with a monobody[15] (Supplementary Fig. 6c). We have now identified the direct interaction between ZapA and the FtsZ N-terminal tail that forms straight protofilaments. Based on the finding, we propose a model of FtsZ conformational equilibrium regulated by ZapA (Fig. 6b). In this model, when FtsZ protofilaments are captured by ZapA molecules, the conformation equilibrium shifts toward FtsZ straight protofilament bundles. During treadmilling of FtsZ protofilaments, the released GDP-bound FtsZ molecules and remaining GDP in the cells still favor the monomeric or curved FtsZ protofilament conformation. However, the interactions with ZapA through the FtsZ N-terminal tail may prevent such a backward shift of the equilibrium to stabilize the more aligned bundled protofilaments for the promotion of cell division. Recent studies have suggested a "cytomotive switch" model in which the conformational switch of FtsZ is induced by polymerization and depolymerization[26–28], rather than GDP/GTP. In addition to the model, a hierarchical model in which interactions with FtsZ binding proteins,

such as ZapA, contribute to biasing a FtsZ conformational equilibrium is conceivable.

We also discuss a mechanistic insight into the cooperative interactions between ZapA and FtsZ. In our HS-AFM analysis, cooperative binding of ZapA to FtsZ filaments was observed at the single-molecule level. This is consistent with the previous experiments by total internal reflection fluorescence microscopy (TIRFM) showing a high cooperativity with a Hill coefficient much greater than 1[17]. Our cryo-EM analysis of the ZapA-FtsZ complex shows no direct interactions between ZapA tetramers. Therefore, the cooperativity is likely caused by structural changes in FtsZ protofilaments. Indeed, we found such structural changes upon ZapA binding to make the adjacent FtsZ molecule more accessible for the next ZapA molecule binding. The interactions of Arg16 of ZapA with Thr52 and Val54 of FtsZ (chain A), those of Glu6 of FtsZ (chain B) and Arg50 of FtsZ (chain A), and those of Thr201 of FtsZ (chain B) and Gln47 of FtsZ (chain A) appear to cause an approximately 2–3 Å shift of β2-α2 loop (indicated by a red arrow in Fig. 4). Such interactions likely rearrange the side chains of Asp209 and Phe210 of FtsZ (chain A), thereby shifting β-sheet comprising of β1–5, as indicated by a blue arrow in Fig. 4. This shift, in turn, causes the shift of α1-β2 loop (residues 31–36) (indicated by a green arrow in Fig. 4), in which Arg33 in the adjacent FtsZ molecule interacts with Gly12 of ZapA We consider that such extensive interactions between ZapA and FtsZ are key to increase positive cooperativity.

Previous TIRFM experiments have suggested that ZapA binds to FtsZ protofilaments without affecting FtsZ treadmilling velocity[17]. The interaction between two FtsZ protofilaments increases electrostatic repulsion, leading to almost no gain in ΔG of dissociation (−3.6 kcal/mol for chain B and chains E and F) (Supplementary Table 2), which is considered essential to form a coherent Z-ring. The increased electrostatic repulsion between protofilaments may explain how closely bundled FtsZ protofilaments can treadmill without interfering with each other. Additionally, the dissociation of ZapA from the ZapA-FtsZ complex may cause FtsZ protofilaments to slide due to repulsive forces, potentially promoting the constriction process.

How can we explain the treadmilling dynamics of the Z-ring if the FtsZ molecules are assembled into double antiparallel protofilaments as observed in the cryo-EM structure of the ZapA-FtsZ complex? Recent high-resolution in vivo fluorescence imaging of *B. subtilis* cells has reported that treadmilling of FtsZ protofilaments is not unidirectional but a mixture of both directions[4]. Especially in nascent Z-rings, occurrences of antiparallel protofilaments colliding to cause temporary pauses in treadmilling were frequently observed. Because ZapA bundles FtsZ protofilaments to stabilize nascent Z-rings[11,17], FtsZ protofilaments are likely bundled in a way that includes both parallel and antiparallel orientations in vivo, as observed in the ZapA-FtsZ complex structure. However, in mature/early contracting Z-rings, the number of pausing FtsZ protofilaments was observed to be reduced, and the average treadmilling velocity was increased[4]. Therefore, we cannot exclude the possibility that other divisome proteins regulate the

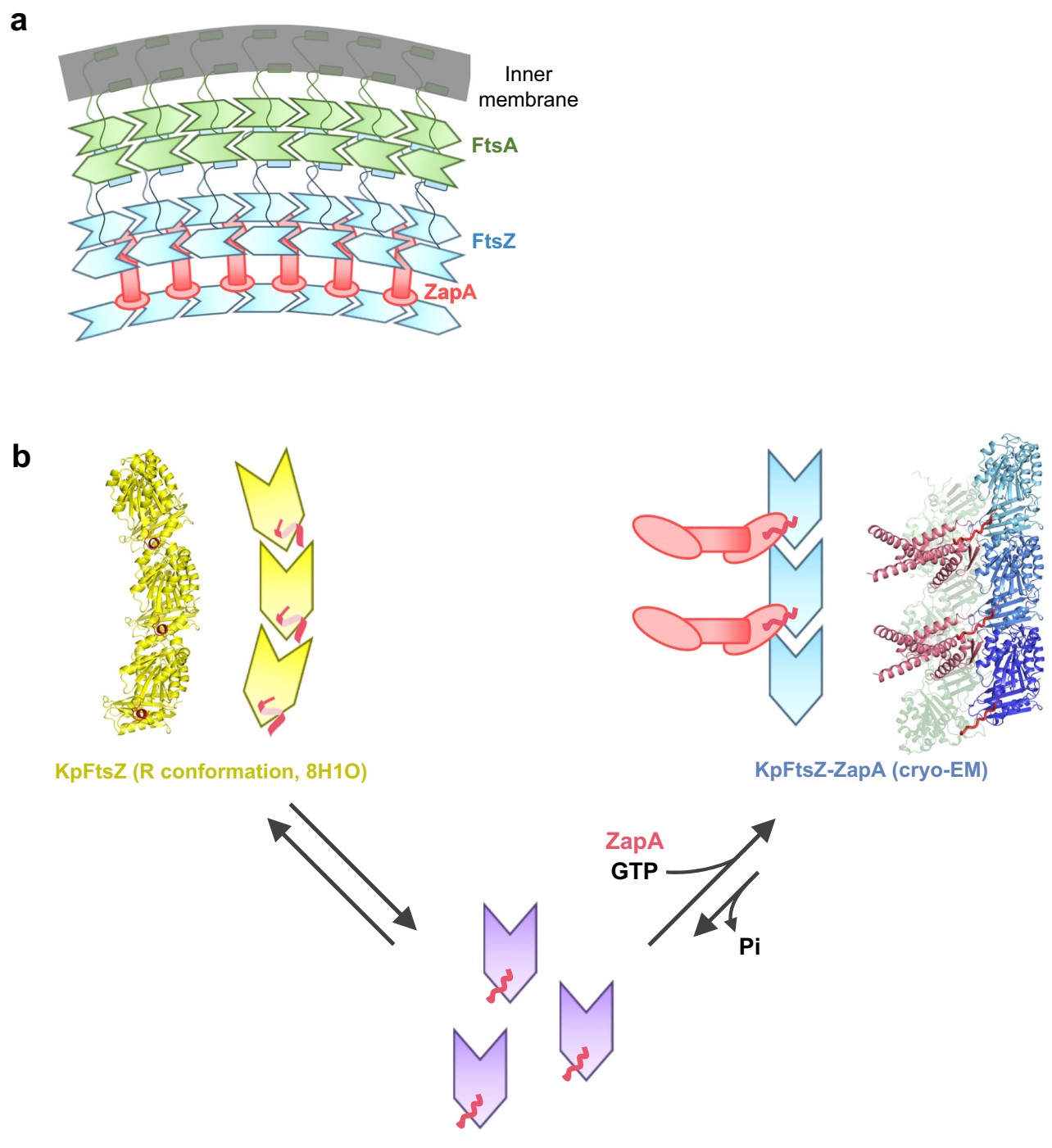

**Fig. 6 | Assembly model of the ZapA-FtsZ. a** A model of interactions between FtsA and the ZapA-FtsZ complex. Double antiparallel FtsZ protofilaments stabilized by ZapA tetramers can bind to double antiparallel FtsA filaments activated by FtsN through the flexible C-terminal linker. **b** FtsZ conformational equilibrium model regulated by ZapA. The N-terminal tail of FtsZ (magenta) adopts a short helix in GDP-bound helical FtsZ protofilament, and the monomeric form of FtsZ must adopt a flexible N-terminal tail, while the tail is captured by ZapA to be extended in the ZapA-FtsZ complex.

orientation and position of the FtsZ protofilaments depending on the stage of cell division.

## Methods
### Protein expression and purification
*K. pneumoniae* FtsZ (KpFtsZ) was expressed and purified as we described previously. A *K. pneumoniae* ZapA-coding gene codon-optimized for *E. coli* was synthesized (Integrated DNA Technologies)

and subcloned into the pColdI-TEV vector (pColdI with a TEV cleavage site) using the NEBuilder HiFi DNA Assembly Cloning Kit (New England Biolabs). The KpZapA_I83E and KpFtsZ_F2A expression vectors were generated using the KOD-Plus Mutagenesis Kit (TOYOBO) by site-directed mutagenesis of the pColdI-TEV-KpZapA vector, and the full-length KpFtsZ vector, which was constructed in our previous study[15]. The sequences of synthesized genes and primers used for genetic manipulations are provided in Supplementary Table 5.

The *E. coli* cells overexpressed KpZapA with N-terminal to His$_6$ tag and a TEV cleavage site were harvested at 4 °C, washed once with buffer A (50 mM Tris-HCl pH 7.5, 300 mM NaCl), and broken by ultrasonication on ice. After ultracentrifugation, the soluble fraction was applied to a 5 ml HisTrap HP column (Cytiva). Elution was carried out with a gradient of 45–500 mM imidazole in buffer A. Peak fractions were collected, and His$_6$ tag was removed with 0.2 µM His-tagged TEV protease while dialyzing overnight against buffer A. His$_6$ tag and His-tagged TEV protease were removed by passing through a second 5 ml HisTrap HP column. Protein fractions were pooled and concentrated on a Vivaspin 20 (MWCO; 5000, Sartorius) to 4 ml and loaded onto a HiLoad 16/600 Superdex200 size-exclusion column (120 ml, Cytiva) equilibrated in buffer B (20 mM HEPES-NaOH pH 7.5, 150 mM NaCl).

### Negative staining

Carbon sides of amorphous carbon grids were glow-discharged by using a JEC-3000FC sputter coater (JEOL). 3 µl of purified ZapA-FtsZ complex at a concentration of 1.0 mg ml$^{-1}$ was loaded on each grid and blotted. Then each grid was immediately stained with 3 µl of 2% uranyl acetate solution and blotted, and this process was repeated three times. Each grid was air-dried for 30 min. Images were taken using JEM-1400Flash (JEOL, Japan) operated at 100 kV.

### Cryo-EM specimen preparation and data collection

The solution containing 12.3 µM (0.5 mg ml$^{-1}$) KpFtsZ, 25.0 µM (0.32 mg ml$^{-1}$) KpZapA, 1 mM GMPCPP in the buffer containing 16 mM HEPES pH 7.5, 14 mM NaCl, 100 mM KCl, and 5 mM MgCl$_2$ were first prepared. This solution was diluted with the same buffer to a final concentration of the solution containing 7.4 µM (0.3 mg ml$^{-1}$) KpFtsZ, 14.9 µM (0.19 mg ml$^{-1}$) KpZapA (2× molar excess to FtsZ), 0.6 mM GMPCPP, 17 mM HEPES pH 7.5, 8.6 mM NaCl, 60 mM KCl, and 3 mM MgCl$_2$, and the diluted sample solution was incubated on ice for 15 min before freezing. Quantifoil grids (R1.2/1.3 Cu 200 mesh) were glow-discharged using a JEC-3000FC sputter coater (JEOL, Japan) at 20 mA for 20 s. 3 µl of the diluted sample solution was applied to the glow-discharged grids in a Vitrobot Mark IV chamber (Thermo Fisher Scientific, USA) equilibrated at 4 °C and 100% humidity. The grids were blotted with a force of −10 and a time of 1.5 s and then immediately plunged into liquid ethane. Excess ethane was removed with filter paper, and the grids were stored in liquid nitrogen. Cryo-EM image datasets were acquired using SerialEM ver. 4.0[29], yoneoLocr ver. 1.0[30], and JEM-Z300FSC (CRYO ARM™ 300: JEOL, Japan) operated at 300 kV with a K3 direct electron detector (Gatan, Inc.) in CDS mode. The Ω-type in-column energy filter was operated with a slit width of 20 eV for zero-loss imaging. The nominal magnification was 60,000×, corresponding to a pixel size of -0.87 Å. Defocus varied between −0.5 µm and −2.0 µm. Each movie was fractionated into 60 frames (0.038 s each, total exposure: 2.29 s) with a total dose of 60 e⁻/Å².

### Cryo-EM image processing and model building

The gain reference was generated from 500 movies in the dataset with the "relion_estimate_gain" program in RELION 4.0[31]. The images were processed using cryoSPARC ver. 4.2.1[32]. To analyze the ZapA-FtsZ complex, a dataset of 7375 movies was imported and motion corrected, the contrast transfer functions (CTFs) were estimated, and 6070 micrographs with the CTF max beyond 5 Å resolution were selected. To prepare a 2D template, 35,635 particles were automatically picked from 500 micrographs using a filament tracer with the following parameters: filament diameter, 200 Å; separation distance between segments, 0.5; minimum filament diameter, 150; maximum filament diameter, 220. The particles were extracted with a box size of 600 pixels with 2x binning. After two rounds of 2D classification, 2D class averages of ladder-like structures located near the center of the box were selected as templates. A total of 4,374,348 particles were automatically picked from all micrographs using the templates in

filament tracer with the following parameters; filament diameter, 44 Å; separation distance between segments, 1; minimum filament length to consider, 5; angular sampling, 0.5°; standard deviation of gaussian blur, 0.4; hysteresis low threshold, 80; radius around crossings to ignore, 0.5. The particles were extracted with a box size of 600 pixels with 2x binning. After two rounds of 2D classification, 565,996 particles were selected and extracted with a box size of 480 pixels binning to 300 pixels.

The extracted particles were subjected to ab initio reconstruction with four classes. To improve the resolution of the FtsZ single protofilament region, we tried many things, including re-centering of the map and local refinement, but all attempts were not successful. Therefore, hereafter we focused on the region of FtsZ double protofilament bound to ZapA tetramer. A total of 148,091 particles were selected as a class showing FtsZ double protofilament and ZapA tetramer, and subjected to another ab initio reconstruction with a single class and maximum resolution of 6 Å. After symmetry search utility, helix refine was performed with an estimated helical rise of 44.67 Å and a helical twist of −1.965°. Particle subtraction was conducted to subtract the signals other than those of the FtsZ double protofilament and ZapA tetramer, and the subtracted particles were subjected to another helix refine with the reversed mask. The generated volume, mask, and particles were aligned with C2 symmetry using volume alignment tools, and C2 symmetry expansion was performed. The symmetry-expanded particles were classified into four classes with 3D classification, and the selected particles were subjected to the removal of duplicate particles. After the map of the best 3D class was aligned back to the original axis (the helical axis corresponding to *z*-axis), helix refine, global and local CTF refine, and another helix refine were performed. Then the particles were subjected to local motion correction with an extracted box size of 360 pixels and a binning to 256 pixels, corresponding to a pixel size of 1.235 Å. The extracted particles were subjected to 2D classification, and 54,670 particles were selected. The selected particles were extracted with a box size of 320 pixels without binning and subjected to another round of helix refine with D1 symmetry, which means a single C2 axis perpendicular to the helical axis but no rotational symmetry along the helical axis[20]. After another round of global and local CTF refine and final helix refine with D1 symmetry, the map resolution reached 2.73 Å (FSC = 0.143) with an optimized helical rise of 44.58 Å and a helical twist of −3.11°. The entire workflow is shown in Supplementary Fig. 1b.

Next, we re-analyzed the same cryo-EM data focusing on ZapA (Supplementary Fig. 8). To obtain the structural information on the single protofilament side, we first tried 3D analysis, including 3D classification, 3D variability, and a similar approach for the ZapA-FtsZ complex reconstruction. Although we attempted multiple rounds of 2D classification focused on the single filament, none were successful. Accordingly, we focused on the region of ZapA tetramer with the FtsZ single protofilament. The particles composing the final ZapA-FtsZ complex model were re-extracted with a box size of 512 pixels with 2x binning, after centering on ZapA tetramer using volume alignment tool. After 2D classification, 207,416 particles were automatically picked using Topaz Train ver. 0.2.5[33] and extracted with a box size of 420 pixels without binning. The extracted particles were subjected to ab initio reconstruction and helix refine with five classes, followed by discarding one class identified as junk. To focus on the ZapA tetramer, the selected 196,074 particles were re-extracted with a smaller box size of 320 pixels binning to 240 pixels and then subjected to 2D classification. The selected 174,332 particles were subjected to helix refine with an estimated helical rise of 44.67 Å and a helical twist of −3.00°, resulting in a 3.89 Å (FSC = 0.143) resolution map. To define the FtsZ single protofilament region more clearly, after re-centering on the ZapA tetramer using volume alignment tool, 60,247 particles were re-extracted with a box size of 420 pixels without binning, and subjected to another round of ab initio reconstruction and helix refine, resulting

in a 4.27 Å (FSC = 0.143) resolution map. The entire workflow is shown in Supplementary Fig. 8.

To analyze the ZapA(E83I)-FtsZ complex, a dataset of 5285 movies was imported, motion corrected, and CTFs were estimated. A total of 5074 micrographs with CTF max beyond 5 Å resolution were selected. To prepare a 2D template, 230,397 particles were automatically picked from 959 micrographs using a manually picked crude 2D reference in the template picker. The particles were extracted with a box size of 256 pixels with 4x binning. After 2D classification, 2D class averages of FtsZ single protofilament were selected as templates. At this step, FtsZ single protofilaments were only observed. A total of 3,409,185 particles were automatically picked from all micrographs using the templates in filament tracer with the following parameters: filament diameter, 44 Å; separation distance between segments, 1; minimum filament length to consider, 4; and angular sampling, 0.5°. The particles were extracted with a box size of 320 pixels with 4x binning. After two rounds of 2D classification, 1,222,245 particles were selected and extracted with a box size of 320 pixels binning to 240 pixels. To remove the bent or ZapA-lacking filament and to sort the filaments by the secondary structure pattern of FtsZ, the extracted particles were subjected to multiple rounds of 2D classification. The entire workflow is shown in Supplementary Fig. 10.

The model of the ZapA-FtsZ complex was built using the cryo-EM structure of the KpFtsZ single protofilament (PDB code: 8IBN) and the crystal structure of ZapA from *E. coli* (PDB code: 4P1M[18]) as initial models. After the initial models were manually fitted into the map using UCSF Chimera ver. 1.16[34] and modified in Coot ver. 0.9.6[35], real-space refinement was performed in PHENIX ver. 1.19.2[36]. The model was validated using MolProbity ver. 4.5.2[37] in PHENIX ver. 1.19. 2[36], and this cycle was repeated several times. The final model contains six FtsZ monomers (three monomers in a single protofilament) and two ZapA tetramers. The cryo-EM data statistics are shown in Supplementary Table 1.

Protein-protein interactions were analyzed with PISA server[21]. Figures were prepared using ImageJ ver. 1.53q[38], UCSF Chimera ver. 1.16[34], ChimeraX ver. 1.6.1[39], CLUSTALX ver. 2.0[40], ESPript ver. 3.0[41] (https://espript.ibcp.fr), LigPlot+ v.2.2[42], and PyMOL ver. 2.5.0 (Schrödinger, LLC, USA).

### Sedimentation assay of FtsZ protofilaments

To investigate the interaction of FtsZ with ZapA, a sedimentation assay was conducted as described previously[43] with minor modifications. 5 μl of protein solution was added to 43 μl of polymerization buffer (50 mM HEPES-KOH (pH 7.5), 100 mM KCl, 5 mM MgCl$_2$). The final concentration of the FtsZ and ZapA is 10 μM, respectively. The solutions were preincubated at 30 °C, 200 rpm for 2 min. 2 μl of 50 mM GTP was added, and the mixture was shaken at 30 °C, 200 rpm for 5 min. Then, the mixture was centrifuged at 4 °C, 20,000×*g* for 19 min. The protein in the pellet and supernatant was assayed by SDS-PAGE.

### Protein crystallization, crystallographic data collection, processing, and refinement

Purified KpZapA at a concentration of 10 mg ml$^{-1}$ was crystallized by hanging-drop vapor diffusion at 293 K (1 μl protein solution + 1 μl reservoir solution) with the reservoir solution consisting of 0.2 M Sodium acetate trihydrate, 0.1 M Sodium citrate pH 5.5, and 10% PEG4000. Crystals were flash-cooled in a stream of nitrogen at 100 K without cryoprotectants after mounting in a loop. X-ray diffraction data were collected at a wavelength of 0.900 Å on the micro-focus beamline BL41XU at SPring-8, Hyogo, Japan, using an EIGER X 16 M detector (Dectris). The datasets were integrated and scaled using the KAMO system[44], which runs BLEND[45], XDS ver. 5 (February 2021)[46], and XSCALE ver. 5 (February 2021)[46] automatically. The phases for each structure were determined by molecular replacement with MOLREP in the CCP4 suite ver. 7.1[47] using the previously determined structure of

EcZapA (PDB code: 4P1M[18]) as the search model. Each model was refined with REFMAC ver. 5.8.0267[48] and PHENIX ver. 1.19.2[36], with manual modification using Coot ver. 0.8.6[35]. The refined structures were validated with MolProbity ver. 4.5.2[37]. Data collection and refinement statistics are shown in Supplementary Table 3.

### HS-AFM measurement and data analysis

HS-AFM observations were carried out using a laboratory-built system. The operating mode of the HS-AFM is the so-called tapping mode, in which the mechanical interaction between the needle probe and the sample is detected by vibrating the cantilever close to its resonant frequency and bringing it into intermittent contact with the sample[49]. The cantilever for the HS-AFM is an Olympus AC10 with a resonant frequency in water of ~500 kHz and a spring constant of ~0.1 N/m. As no sharp probes were designed for the tip of the HS-AFM cantilever, an amorphous carbon probe was fabricated by electron beam deposition and further sharpened by plasma etching in an Ar atmosphere to obtain a probe with a tip radius of curvature of less than about 5 nm[50].

For HS-AFM observations of ZapA-FtsZ complexes, two approaches were used: either pre-mixing FtsZ and ZapA in a tube before adsorption onto a mica substrate (referred to as "pre-mixing method" in figure legends), or first adsorbing only FtsZ onto the mica substrate, followed by the addition of ZapA to the observation solution (referred to as "ZapA addition method" in figure legends). For the former approach, following the same procedure used for cryo-EM analysis, 14 μM KpFtsZ, 36 μM KpZapA, and 1 mM GTP were mixed in buffer solution (20 mM HEPES-NaOH pH 7.5, 5 mM MgCl$_2$, 100 mM KCl) and incubated for 5 min before being adsorbed onto a freshly cleaved mica substrate. After 10 min of incubation, unbound molecules were removed by washing with the same buffer. HS-AFM observations were then performed in buffer solution containing nucleotides. For the latter approach, FtsZ filaments were adsorbed onto a mica substrate under the aforementioned conditions without KpZapA, followed by observations in buffer solution containing nucleotides, and then ZapA was added to the buffer solution to observe the interaction between ZapA and FtsZ filaments. As nucleotide conditions and ZapA concentrations varied between sessions, they are individually specified in the figure legends. All measurements were performed at room temperature. The HS-AFM images shown in the manuscript were processed with tilt compensation and smoothing filters to reduce the image noise. The analysis of the dynamic interaction of ZapA on FtsZ protofilaments was performed by measuring the time from binding to dissociation for isolated ZapA bright spots on the images, as far as possible avoiding spots with neighboring molecules, and creating histograms based on this data. Image filtering and analysis of HS-AFM images were performed using custom image processing software developed with Igor Pro 9.0 (WaveMetrics Inc., Lake Oswego, OR).

### Reporting summary

Further information on research design is available in the Nature Portfolio Reporting Summary linked to this article.

## Data availability

The cryo-EM atomic coordinates and maps of *K. pneumoniae* FtsZ double filament-ZapA tetramer have been deposited in the Protein Data Bank (PDB) and the Electron Microscopy Data Bank (EMDB) under accession codes 9ISK and EMD-60837. The coordinates and structure factors of *K. pneumoniae* ZapA have been deposited in PDB under the accession number 9ISJ. The following previously published PDB codes are referred to: 1W2E; 7OHK; 4P1M; and 8IBN. Correspondence and requests for materials should be addressed to Takayuki Uchihashi or Hiroyoshi Matsumura. Source data are provided with this paper.

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

## Acknowledgements

We thank Yoshie Kushima, Reiko Yamauchi, and Takamoto Konishi for their help with negative staining and preliminary experiments. This work was supported by: JSPS KAKENHI grant JP20K22630 (J.F.), JP23K06418 (R.U.), JP24K01994 (H.M.), JP24H02277 (H.M.), JP24H02270 (H.M.), JP23K18033 (H.M.), JP25H02292 (H.M.), JP24K0130 (T.U.); MEXT Promotion of Development of a Joint Usage/ Research System Project: Coalition of Universities for Research Excellence Program (CURE) (Grant Number JPMXP1323015482) (T.U.); Uehara Memorial Foundation (H.M.); Nagase Science and Technology Foundation (H.M.); The NOVARTIS Foundation (Japan) for the Promotion of Science (H.M.); Naito Science & Engineering Foundation (H.M.), G-7 Foundation (R.U.); JST OPERA (Open Innovation with Enterprises, Research Institute and Academia) grant JPMJOP1861 (K.N.); the Program for the R-GIRO Research from the Ritsumeikan Global Innovation Research Organization, Ritsumeikan University (H.M.); AMED BINDS (Platform Project for Supporting Drug Discovery and Life Science Research (BINDS)) grant JP21am0101117 and JP22ama121003 (K.N.), JP24ama121003 (K.N.), and JP23ama121001 (H.M.); AMED CiCLE (Cyclic Innovation for Clinical Empowerment) grant JP17pc0101020 (K.N.); JEOL YOKOGUSHI Research Alliance Laboratories of The University of Osaka (K.N.); the Cooperative Research Program of the Institute for Protein Research, The University of Osaka (CR-22-02 and CR-23-02).

## Author contributions

J.F. and H.M. designed the research. K.H., N.K., S.T., Y.K. and R.U. prepared the protein sample and performed crystallization, crystallographic data collection, processing, refinement, and model building. J.F., K.H. and N.K. performed negative staining and cryo-EM data collection. J.F. and K.K. performed cryo-EM image processing and model building. G.K. and T.U. performed HS-AFM experiments and analyzed the data. J.F., K.K., T.U. and H.M. prepared the figures and wrote the first draft of the manuscript. R.U. and K.N. helped to analyze and interpret the data and critically revise the manuscript. J.F. and H.M. conceptualized the study, developed the study design, supervised the authors throughout the study, and provided expertise in manuscript preparation. All authors read and approved the reviewed manuscript.

## Competing interests

The authors declare no competing interests.
