## [Peer Review file · Nature Communications]

Structural basis for the interaction between the bacterial cell division proteins FtsZ and ZapA

Corresponding Author: Professor Hiroyoshi Matsumura

Version 0:

Reviewer comments:

Reviewer #1

(Remarks to the Author)

“Structural basis for the interaction between the bacterial cell division proteins FtsZ and ZapA”

The authors used cryo-electron microscopy and High-speed atomic force microscopy to reveal and analyze the structure and characteristics of the structure of the FtsZ-ZapA complex. These studies are straightforward, high quality of the data, the paper is well presented, and have benefits for understanding the structure of the Z-ring and the mechanism of bacterial division. However, some results and conclusions are worth discussing.

1. One of the main points of the authors is that the FtsZ-ZapA complex formed by FtsZ, ZapA and GMPCPP is an asymmetric structure, in which the double antiparallel FtsZ protofilament on one side and a single protofilament on the other side are tethered by ZapA tetramers, and they believed that this has physiological functions in vivo. But how to explain the treadmilling dynamics of the Z-ring if FtsZ filaments are double antiparallel? The treadmilling are more likely to occur in a parallel double filament or a single filament. Is this antiparallel double protofilament just an artifact? Because the FtsZ protofilament induced by GMPCPP will form a bundle structure, this bundle structure may be formed only due to the charges distributed around the protofilaments, similar to the formation of the antiparallel one.

2. The authors observe the structures of ZapA bridging two single FtsZ protofilaments using HS-AFM. In this experiment, authors used FtsZ, ZapA and GTP, and thought this might be an artifact. But I think this may be real, because the authors used GTP here. So these figures and results may be important and should be included in the figures.

3. In subsequent experiments, the authors preadsorbed FtsZ protofilament, then added ZapA and observed ZapA binding to the hollow sites between two FtsZ protofilaments. I wonder if this experiment will bring artificial results? Since the preadsorbed FtsZ protofilaments may not be able to move freely, ZapA may not be completely bound to FtsZ due to the distance between the two FtsZ protofilaments being too close or too far, so binding and dissociation of ZapA are observed in experiments. In previous studies, the dynamic effect of FtsZ-ZapA is that the binding of ZapA does not affect the GTP hydrolysis activity of FtsZ, and the disassembly of FtsZ protofilament.

4. In previous studies, it was found that the bundle structures formed by FtsZ-ZapA are straight, which may be inconsistent with the curved structure of FtsZ-ZapA in the model.

Reviewer #2

(Remarks to the Author)

In this manuscript, the authors describe the cryo-EM structure of a complex formed by FtsZ filaments and ZapA from *K. pneumoniae*. The structure shows how an elongated ZapA tetramer perpendicularly bundles two antiparallel FtsZ filaments on one side and a single filament on the other. Two consecutive FtsZ subunits in each protofilament contact the ZapA tetramer with minor conformational changes in both components, as compared to the ZapA crystal structure included in the manuscript and a former cryo-EM structure of the FtsZ protofilament by the authors' laboratory (Fujita et al., Nat Comm

2023). Structural work is complemented by HS-AFM data showing that ZapA binding to FtsZ protofilaments is dynamic and presents cooperativity.

This is an interesting study with potential biological significance in the field of bacterial cell division. The manuscript is well-written, experimental approaches are thoroughly described and the figures effectively illustrate the key findings. However, there are several shortcomings in the manuscript that need to be addressed prior to publication.

My main concern relates to the observed asymmetry in the number of protofilaments on each side of the ZapA tetramer. If the two heads in the ZapA tetramer are identical, then a similar arrangement can be expected on both sides. Moreover, the authors claim that the asymmetric arrangement is not observed in their HS-AFM data (lines 176-178). Can they explain this discrepancy? Moreover, crystal structures ZapA from *K. pneumoniae* (this work) and *E. coli* (Low et al., JMB 2004) show that the two ZapA heads are related by a $\sim 90^\circ$ turn around an axis that runs along the ZapA coiled coil. Superposition of the cryo-EM FtsZ-ZapA structure on both ZapA heads would result in two perpendicular bundles of double FtsZ protofilaments. This can be relevant in vivo, as it may facilitate adaptation to a curved membrane surface. There are two options here. The authors can prepare a ZapA mutant only able to form dimers and not tetramers, and apply cryo-EM to assess whether this mutant binds single and double FtsZ filaments or only double filaments. Alternatively, the authors can prepare grids with thicker ice and collect cryo-electron tomograms. This approach would reduce the effect of the thin ice layer formed in the holes of cryo-EM grids upon blotting, which likely induces torsion in the ZapA coiled coil, making FtsZ filaments on both sides of the ZapA tetramer to adopt a roughly parallel configuration, as seen in Ext. Data Fig. 2a.

The authors should also clarify why they were unable to resolve the structure of the single FtsZ protofilament on one side of the complex. They claim that they performed “many trials” (line 93) to improve resolution in this region. However, it remains unclear whether they tried to mask away and subtract the signal from the double protofilament and half of the ZapA tetramer to reconstruct the single protofilament, followed by 3D classification. They should also attempt other tools (3DFlex, variability analysis...) to assess putative conformational heterogeneity for this part of the assembly.

An interesting finding is the fact that ZapA-mediated double protofilaments run antiparallel to each other. Can the authors provide evidence that this can actually happen in vivo? This would enrich the discussion. As example, double FtsZ protofilaments that were ~ 6.8 nm apart have been observed in cryo-ET studies of *E. coli* (Szwedziak et al., eLife 2014). How does this relate to the current results? Moreover, the potential implications of the observed electrostatic repulsion in the FtsZ double protofilaments for FtsZ dynamics should be further elaborated.

While structural data are robust, additional functional assays would support the proposed mechanisms. For this, the authors should prepare mutants and evaluate their effects in vitro or in vivo. Among mutants, they could consider deletion of the FtsZ N-terminal tail, as well as preparing point mutants at other interacting surfaces (FtsZ-ZapA & FtsZ-FtsZ). For instance, to evaluate whether electrostatic repulsion of FtsZ filaments is relevant for treadmilling, mutants E238R or E305R can be prepared.

Upon inspection of the cryo-EM maps and models, I am not fully convinced by the placement of GMPCPP in two conformations. Superposition of the crystal structure of FtsZ from *S. aureus* in complex with GDP, BeF₃⁻ and Mg²⁺ (Ruiz et al., PLoS Biol 2022) onto the current filament shows that Mg²⁺ (and its water coordination sphere) nicely fits the density that is attributed to the GMPCPP gamma-phosphate in conformation 2. Have the authors considered the possibility of modelling only conformation 1 for GMPCPP and adding Mg²⁺ in the density currently attributed to the gamma-phosphate in conformation 2? Besides, Fig. 3b should include a superposition with 7OHK (SaFtsZ with GDP, BeF₃⁻ and Mg²⁺) rather than 3WGN (SaFtsZ with GTPgammaS), as the former is a better mimetic of GTP function.

Regarding HS-AFM experiments, the authors should clarify how they distinguish single from double filaments in the crowded images shown in Fig. 5. They likely rely on height measurements to differentiate between single and double filaments, but height quantitative data are missing. To strengthen this claim, the authors should provide quantitative height profile analyses and statistical analysis of filament widths and heights. Importantly, experiments shown in Fig. 5 were performed in the presence of GMPPNP (lines 398-399), which the authors' laboratory has showed to form tubes instead of protofilaments (see Fig. 1 in Fujita et al., Nat Comm 2023). This raises the question of whether the filaments shown in current Fig. 5 are protofilaments in the straight configuration, as claimed by the authors (line 401), or tubes. In the latter case, the description and conclusions should be revisited.

Minor points:

- The authors should consider using “dimer of dimers” instead of “tetramers” for ZapA, as there is no 4-fold axis relating the four protomers
- Line 148. It should read “and similar conformations” rather than “and the similar conformations”
- Lines 166-167. Please, quantify shifts in beta-sheet and T7 loop. Please, describe how this affects the cleft aperture (if at all) and/or GTPase activity (if possible to estimate)
- Line 228. Reference should be “Nierhaus et al., Nat Microb 2022”
- Line 267. Full-stop missing
- Ext. Data Fig. 5. Please, use original nomenclature for FtsZ helices and strands (Nogales et al., NSMB 1998)
- Ext. Data Table 3. Please, report ligand nature, as in Ext. Data Table 1

(Remarks to the Author)

The authors employed cryogenic electro-microscopy (cryo-EM) and high-speed atomic force microscopy (HS-AFM) to investigate the structure and dynamics of FtsZ-ZapA complex. For the cryo-EM study, the authors resolved an asymmetric ladder-like structure, where ZapA tetramers tethered a double antiparallel FtsZ protofilament on one side and a single protofilament on the other. Furthermore, the authors elaborated on detailed structural features underlying interactions between the double protofilament and the interactions between FtsZ and ZapA molecules, as well as calculated the free energy to support their conclusions. Using HS-AFM, the authors explored the single-molecule dynamics of the ZapA-FtsZ complex, employing two approaches: first, by pre-mixing ZapA, FtsZ, and GTP in a tube prior to HS-AFM imaging, and second, by pre-absorbing FtsZ protofilaments onto a mica substrate and subsequently adding ZapA to the solution for HS-AFM imaging.

I was particularly invited to technically assess the HS-AFM data and conclusions drawn. While the HS-AFM data is of high quality, with clearly resolved single molecules, I find the interpretation of these results less convincing, and believe that further experiments, data analysis, or clarifications are necessary.

To me, the cryo-EM data clearly indicated an asymmetric ladder-like structure composed of a double protofilament on the one side and a single protofilament on the other side. However, like the authors concluded which I agree, the HS-AFM study and cross-sectional height analysis in Extended Data Fig.7, exclusively show ladder-like structures where ZapA bridges two single FtsZ protofilaments. I think this discrepancy is indeed surprising and intriguing, but I am not fully convinced that this observation is an artifact as the authors concluded. Here are my concerns:

1. Did the authors attempt to dilute their pre-mixture samples before mica absorption? It seems that Extended Data Fig.7 suggests a dense molecular distribution on the mica, whereas the ZapA-FtsZ complex likely requires more space for proper absorption.
2. As far as I observed, many filaments in Extended Data Fig. 7 appear similar to those FtsZ protofilaments in Fig 5a-d where, as the authors noted, the absorption surface of FtsZ molecules to the mica is opposite to the ZapA binding surface. If this is the case, and ZapA, FtsZ, and GTP were pre-mixed and given sufficient time to form complexes before HS-AFM sample application, why didn't the authors observe bright dots (ZapA bound to FtsZ) in Extended Data Fig. 7?
3. I am not convinced that the distortion upon absorption onto the planar substrates disrupted ZapA-mediated cross-linking between double protofilaments and single protofilaments and converted them to cross-linking of single protofilaments. The ladder-like structures in the HS-AFM experiments (Extended Data Fig. 7) appear too neat, orderly, and clean to result from such disruptions. If the authors' interpretation is correct, I would expect to observe short, irregular or even broken structures of tethered single protofilaments
4. The dwell time analysis in Fig.5, indicates that ZapA interacts more strongly with the FtsZ double protofilament than the single protofilament. Despite the authors's argument that the inter-protofilament interactions with the double FtsZ filaments is weak, I still find it unlikely that a conversion from double to single protofilaments, which requires breaking stronger ZapA interactions with two FtsZ and subsequent formation of new weaker interactions with single FtsZ protofilament, would result in the neat structures observed in Extended Data Fig.7.
5. It might be also helpful if the authors provided more stats on these protofilaments from the native complex, or at least more HA-AFM images from different regions (and perhaps additional replicates?).

I think the HS-AFM experiments where the authors first applied FtsZ alone and added ZapA to study the interaction dwell times are well-conceived. The conclusion that the adsorption surface of FtsZ to the mica substrate is opposite to the ZapA binding surface is technically sound. However, I think the data analysis needs refinement to support the claim of increased affinity and provide additional mechanistic insight into the work. This conclusion could be strengthened by further dwell time analysis, discriminating all binding events to double protofilaments into two categories: those involving isolated ZapA molecules and those where ZapA molecules are adjacent. It is apparent in Fig. 5b that the binding events represent a mixture of (at least) these two cases. As a result, the dwell time histogram in panel Fig. 5e (at least the right one) should contain more than one time constant (at least two exponentials). It would also be valuable to explore whether this additional affinity primarily arises from binding to double protofilaments, as this could provide further novel mechanistic insights into the previously observed binding cooperativity.

In summary, while the HS-AFM data quality is high, the interpretation and analysis require improvement to meet the publication standards of Nature Communications. In the current version, I respectfully believe that the HS-AFM observations of increased affinity in Fig.5 do not present significant novelty complementing the cryo-EM study, and beyond what has already been observed in previous studies regarding the cooperative binding. However, with further analysis, these experiments hold substantial potential to provide valuable insights into the structural dynamics of the system. Besides, the observation of single FtsZ protofilaments tethered by ZapA molecules is exciting. While it is premature to conclude that this is an artifact, it certainly warrants further investigation and could introduce novel insights into the study.

Version 1:

Reviewer comments:

Reviewer #1

(Remarks to the Author)

The author answered my questions and I recommend it for publication.

Reviewer #2

(Remarks to the Author)

The authors have substantially improved the manuscript. Notably, they have reanalyzed their FtsZ-ZapA cryo-EM data by focusing on ZapA, which allowed better visualization of the full ZapA tetramer and the single FtsZ protofilament on one side of the ZapA tetramer. They also present a new cryo-EM structure of FtsZ complexed to a ZapA point mutant that renders ZapA dimeric, and show that it binds single FtsZ protofilaments through different arrangements. Additionally, they prepared a mutant at the N-terminus of FtsZ (F2A) that loses binding with ZapA, further stressing the relevance of the interactions observed in their cryo-EM structure. Moreover, they reinterpreted the density for GMPCPP in the FtsZ active site. Finally, their revised discussion better reflects the conclusions derived from the results. Overall, the authors have adequately addressed my concerns.

Reviewer #3

(Remarks to the Author)

In their revised manuscript, Fujita et al. have conducted additional experiments that effectively address all of my previous concerns. I am pleased to see that these new experiments have led to a stronger conclusion regarding the highly dynamic and fluctuating intermolecular interactions. These insights are made possible by HS-AFM and serve as a valuable complement to the cryo-EM structural analysis (If possible, I suggest that the authors emphasize this point more clearly in the manuscript.)

Overall, the HS-AFM data and analysis presented in this study are of high quality, and I therefore recommend the publication of this work in Nature Communications.

REVIEWER COMMENTS

Reviewer #1 (Remarks to the Author):

The authors used cryo-electron microscopy and High-speed atomic force microscopy to reveal and
analyze the structure and characteristics of the structure of the FtsZ-ZapA complex. These studies are
straightforward, high quality of the data, the paper is well presented, and have benefits for
understanding the structure of the Z-ring and the mechanism of bacterial division. However, some
results and conclusions are worth discussing.

Thank you for your encouraging comments.

1. One of the main points of the authors is that the FtsZ-ZapA complex formed by FtsZ, ZapA and
GMPCPP is an asymmetric structure, in which the double antiparallel FtsZ protofilament on one side
and a single protofilament on the other side are tethered by ZapA tetramers, and they believed that this
has physiological functions *in vivo*. But how to explain the treadmilling dynamics of the Z-ring if FtsZ
filaments are double antiparallel? The treadmilling are more likely to occur in a parallel double
filament or a single filament. Is this antiparallel double protofilament just an artifact?

Re1-1: Recent high-resolution *in vivo* fluorescence imaging of *B. subtilis* cells has reported that
the treadmilling direction of FtsZ protofilaments is not unidirectional but a mixture of both
directions (Whitley, K.D et al., Nat Commun 12, 2448 (2021)). Especially in nascent Z-rings,
occurrences of antiparallel filaments colliding to cause temporary pauses in treadmilling have
been frequently observed. Because ZapA functions to bundle FtsZ protofilaments to stabilize
nascent Z-rings, FtsZ protofilaments are likely bundled in a way that includes both parallel and
antiparallel orientations *in vivo*, as observed in the ZapA-FtsZ complex structure. Thus, we
consider the antiparallel double protofilament to be not an artifact, and we discuss it in lines 335-
346.

Because the FtsZ protofilament induced by GMPCPP will form a bundle structure, this bundle
structure may be formed only due to the charges distributed around the protofilaments, similar to the
formation of the antiparallel one.

Re1-2: Previous electron microscopy observations using negative staining of KpFtsZ (Fig. 3a, b
in Fujita et al., 14, 4073, Nat. Commun. 2023) have shown that KpFtsZ protofilaments are bundled
to the same extent with both GTP or GMPCPP. To confirm the reproducibility, we performed the
negative-stain observation experiment of KpFtsZ-GMPCPP again under the same conditions as in
Fig.3a,b of the previous study, as shown below. We did not observe an increase in FtsZ
protofilament bundling. Therefore, this point about GMPCPP does not seem to apply to KpFtsZ.

2. The authors observe the structures of ZapA bridging two single FtsZ protofilaments using HS-AFM. In this experiment, authors used FtsZ, ZapA and GTP, and thought this might be an artifact. But I think this may be real, because the authors used GTP here. So these figures and results may be important and should be included in the figures.

Re1-3: To respond to this reviewer's comments, we re-analyze the same cryo-EM data of the ZapA-FtsZ complex focusing on ZapA, and also performed a new cryo-EM analysis of the ZapA dimer mutant (I83E)-FtsZ complex. We added the results of these analyses in lines 163-194. The analyses revealed multiple distinct interaction patterns between the ZapA dimer and a single FtsZ protofilament, suggesting that the ZapA-FtsZ interaction is highly dynamic and fluctuating, because of the interaction through the flexible N-terminal tail of FtsZ. We also have to consider the constraints of HS-AFM observations, where molecules are imaged on a two-dimensional mica substrate. Among those interaction patterns, only molecules compatible with binding on the mica surface must be observed in HS-AFM, where one is the single protofilament-ZapA-single protofilament configuration (Fig. 5a-c), and the other is ZapA molecules on double or single protofilaments (Fig. 5e-g). Therefore, we deleted the "artifact" and rewrote the discussion in lines 270-275. Our additional HS-AFM experiments showed no difference in GTP, GMPCPP, and GMPPNP results. We thus moved Extended Data Fig. 7 to Fig. 5a-c.

3. In subsequent experiments, the authors preabsorbed FtsZ protofilament, then added ZapA and observed ZapA binding to the hollow sites between two FtsZ protofilaments. I wonder if this experiment will bring artificial results? Since the pre-adsorbed FtsZ protofilaments may not be able to move freely, ZapA may not be completely bound to FtsZ due to the distance between the two FtsZ protofilaments being too close or too far, so binding and dissociation of ZapA are observed in experiments. In previous studies, the dynamic effect of FtsZ-ZapA is that the binding of ZapA does not affect the GTP hydrolysis activity of FtsZ, and the disassembly of FtsZ protofilament.

Re1-4: We have observed that the FtsZ protofilaments fluctuated somewhat on the mica substrate.
Therefore, we added that movie (Supplementary Video 3) to avoid misleading the reader and
described it in line 228-230 of the Results section.

4. In previous studies, it was found that the bundle structures formed by FtsZ-ZapA are straight, which
may be inconsistent with the curved structure of FtsZ-ZapA in the model.

Re1-5: Negative staining microscopy has shown that the ladder-like structures are flexible; indeed,
we frequently observed twisted ladder-like structures. Since the cryo-EM structure represents an
averaged structure, it likely appears linear locally. We think it would be better not to change Figure
6b for such reasons, as this would be consistent with the current manuscript.

Reviewer #2 (Remarks to the Author):

In this manuscript, the authors describe the cryo-EM structure of a complex formed by FtsZ filaments
and ZapA from *K. pneumoniae*. The structure shows how an elongated ZapA tetramer perpendicularly
bundles two antiparallel FtsZ filaments on one side and a single filament on the other. Two consecutive
FtsZ subunits in each protofilament contact the ZapA tetramer with minor conformational changes in
both components, as compared to the ZapA crystal structure included in the manuscript and a former
cryo-EM structure of the FtsZ protofilament by the authors' laboratory (Fujita et al., Nat Comm 2023).
Structural work is complemented by HS-AFM data showing that ZapA binding to FtsZ protofilaments
is dynamic and presents cooperativity.

This is an interesting study with potential biological significance in the field of bacterial cell division.
The manuscript is well-written, experimental approaches are thoroughly described and the figures
effectively illustrate the key findings. However, there are several shortcomings in the manuscript that
need to be addressed prior to publication.

Thank you for your encouraging comments.

My main concern relates to the observed asymmetry in the number of protofilaments on each side of
the ZapA tetramer. If the two heads in the ZapA tetramer are identical, then a similar arrangement can
be expected on both sides. Moreover, the authors claim that the asymmetric arrangement is not
observed in their HS-AFM data (lines 176-178). Can they explain this discrepancy? Moreover, crystal
structures ZapA from *K. pneumoniae* (this work) and *E. coli* (Low et al., JMB 2004) show that the two
ZapA heads are related by a $\sim 90^\circ$ turn around an axis that runs along the ZapA coiled coil.
Superposition of the cryo-EM FtsZ-ZapA structure on both ZapA heads would result in two
perpendicular bundles of double FtsZ protofilaments. This can be relevant in vivo, as it may facilitate
adaptation to a curved membrane surface.

Re2-1 First, we reply to the first reviewer #2's comment, "similar arrangements can be expected
on both sides". As this reviewer points out, the orientation of the right side ZapA dimer head is
quite different from that of the left side. This was clearly indicated in our ZapA-focused re-analysis
of the cryo-EM data of the ZapA-FtsZ complex (Extended Data Fig. 9). Because of the different
orientation of the ZapA dimer head on both sides, the same interactions of ZapA with FtsZ cannot
occur. However, since the right side dimer head is aligned in a straight line with equal spacing,
ZapA molecules can bind to the single protofilament through a multipoint interaction via the
extended FtsZ N-terminal tail. We add the discussion in lines 276-283.

Next, we would reply to the second comment, "relevant *in vivo*". Recent high-resolution *in vivo*
fluorescence imaging of *B. subtilis* cells has reported that the treadmilling of FtsZ
protofilaments is not unidirectional but a mixture of both directions (Whitley, K.D et al., Nat
Commun 12, 2448 (2021)). Especially in nascent Z-rings, occurrences of antiparallel
protofilaments colliding to cause temporary pauses in treadmilling were frequently observed. We
also performed a re-analysis of the same cryo-EM data of the ZapA-FtsZ complex focusing on
ZapA and a new cryo-EM analysis of the ZapA dimer mutant (I83E)-FtsZ complex in lines 163-
194, as suggested by Reviewer #2 (Extended Data Figs. 8, 9, We appreciate it very much). The
analyses revealed multiple distinct interaction patterns between ZapA and single FtsZ
protofilaments, suggesting that the ZapA-FtsZ interaction is highly dynamic and fluctuating. Such
interactions allow for forming the asymmetric ZapA-FtsZ complex, in which a single
protofilament is tilted compared to the double protofilaments (Extended Data Fig. 2). We think
this molecular feature of ZapA-FtsZ must be necessary for *in vivo* interactions with curved
membranes, as reviewer #2 pointed out. We therefore added the discussion in lines 282-288.

There are two options here. The authors can prepare a ZapA mutant only able to form dimers and not
tetramers, and apply cryo-EM to assess whether this mutant binds single and double FtsZ filaments or
only double filaments. Alternatively, the authors can prepare grids with thicker ice and collect cryo-
electron tomograms. This approach would reduce the effect of the thin ice layer formed in the holes of
cryo-EM grids upon blotting, which likely induces torsion in the ZapA coiled coil, making FtsZ
filaments on both sides of the ZapA tetramer to adopt a roughly parallel configuration, as seen in Ext.
Data Fig. 2a. Alternatively, the authors can prepare grids with thicker ice and collect cryo-electron
tomograms.

Re2-2: Thank you for the reviewer's suggestion. As the reviewer suggested, we performed a cryo-
EM analysis of the ZapA dimer mutant (I83E) complexed with FtsZ, and re-analyzed the same
cryo-EM data of the ZapA-FtsZ complex focusing on ZapA (Extended Data Fig. 8-11). As
described above in Re2-1, we found that the interaction of ZapA with FtsZ single protofilaments
is highly dynamic and fluctuating. Further, recent high-resolution *in vivo* fluorescence imaging of
*B. subtilis* cells has reported that the treadmilling direction is not unidirectional but a mixture of
both directions (Whitley, K.D et al., Nat Commun 12, 2448 (2021)). Especially in nascent Z-rings,

occurrences of antiparallel protofilaments colliding to cause temporary pauses in treadmilling
were frequently observed. Because ZapA bundles FtsZ protofilaments to stabilize nascent Z-rings,
FtsZ protofilaments are likely bundled in a way that includes both parallel and antiparallel
orientations *in vivo*, as observed in the ZapA-FtsZ complex structure. Thus, we consider the
antiparallel double protofilament to be not an artifact. Because this reviewer's point is important,
we added the comments in the discussion section of lines 335-346.

The authors should also clarify why they were unable to resolve the structure of the single FtsZ
protofilament on one side of the complex. They claim that they performed “many trials” (line 93) to
improve resolution in this region. However, it remains unclear whether they tried to mask away and
subtract the signal from the double protofilament and half of the ZapA tetramer to reconstruct the
single protofilament, followed by 3D classification. They should also attempt other tools (3DFlex,
variability analysis...) to assess putative conformational heterogeneity for this part of the assembly.

Re2-3: To respond to the reviewer's comments, we re-analyzed the same cryo-EM data of the
ZapA-FtsZ complex focusing on ZapA (Extended Data Fig. 8, 9). This analysis indicated that the
orientation of the ZapA dimer head on the single protofilament side is quite different from that on
the double protofilament side, emphasizing that the ZapA-FtsZ complex represents an asymmetric
structure. Detailed comments are added in line 163-181 in the Results section. Although we also
performed 3DFlex variability and 3D classification for single protofilaments, none was successful.

An interesting finding is the fact that ZapA-mediated double protofilaments run antiparallel to each
other. Can the authors provide evidence that this can actually happen *in vivo*? This would enrich the
discussion. As example, double FtsZ protofilaments that were ~6.8 nm apart have been observed in
cryo-ET studies of *E. coli* (Szwedziak et al., eLife 2014). How does this relate to the current results?
Moreover, the potential implications of the observed electrostatic repulsion in the FtsZ double
protofilaments for FtsZ dynamics should be further elaborated.

Re2-4: As we describe in Re2-2, the treadmilling direction is not unidirectional (Whitley, K.D et
al., Nat Commun 12, 2448 (2021)). Therefore, FtsZ filaments are likely bundled *in vivo* in a
manner that includes both parallel and antiparallel orientations, consistent with our ZapA-FtsZ
complex analysis.

Regarding the reviewer's latter point, our data do not account for protofilaments spaced 6.8
181 nm apart. However, we cannot exclude the possibility that other divisome proteins regulate the
182 orientation and position of the FtsZ protofilaments depending on the stage of cell division, and we
added a discussion in lines 335-346. We also added a discussion on electrostatic repulsion in lines
332-334 in the Discussion section, arguing that the dissociation of ZapA from the ZapA-FtsZ
complex may cause FtsZ protofilaments to slide due to repulsive forces potentially promoting the
constriction process.

While structural data are robust, additional functional assays would support the proposed mechanisms.
For this, the authors should prepare mutants and evaluate their effects in vitro or in vivo. Among
mutants, they could consider deletion of the FtsZ N-terminal tail, as well as preparing point mutants at
other interacting surfaces (FtsZ-ZapA & FtsZ-FtsZ). For instance, to evaluate whether electrostatic
repulsion of FtsZ filaments is relevant for treadmilling, mutants E238R or E305R can be prepared.

Re2-5: We further analyzed by mutagenesis. We identified Phe2 as the most significant contributor
among the N-terminal residues, and generated the FtsZ-F2A mutant (F2A). Polymerization assays
and HS-AFM observations revealed F2A lost its ability to bind ZapA. Therefore, we prepared
Extended Data Fig. 7, Extended Data Table 4, and Supplementary Videos 1,2. Details are
described in lines 150-160 in the Results section.

Regarding in vivo experiments, we observed EcFtsZ-F2A-mCherry overexpressed E.coli
cells using differential interference contrast (DIC) and fluorescence microscopy as shown below
(in Fig. Re2-5-1). Detailed methods are provided in Supplementary Method R1 for manuscript
review purposes only. Since the amino acid sequence identity of Kp and EcZapA is high and the
amino acid residues involved in the interactions of ZapA with FtsZ are conserved (Extended Data
Fig. 4), we assumed that the *E. coli* ZapA-FtsZ complex structure is essentially the same as that
of *K. pneumoniae*, and performed experiments in *E. coli*. As shown in Fig. 3 in Gueiros-Filho, F.J.
and Losick R. , Gene Dev, 16, 2544 (2002), our observation shows overexpression of FtsZ shows
a multiple Z-ring. However, there is no clear difference in Z-ring morphology between the WT
and F2A mutant-overexpressed cells.

Fig. Re2-5-1: Differential interference contrast (DIC) and fluorescence microscopy images of
EcFtsZ-F2A-mCherry overexpressed E.coli cells using

We then investigated the effect of the EcFtsZ F2A overexpression on the colony formation (Fig.
Re2-5-2). We observed slight differences in the colony formation between the WT and F2A
mutant-overexpressed *E.coli* cells, but we decided it was too subtle to write about in a paper.

Fig. Re2-5-2: Colony formation of EcFtsZ and F2A overexpressed *E. coli* cells

We guess the defects of the interaction of ZapA with FtsZ may be compensated for by other Zap proteins (ZapB, ZapC, ZapD, etc.) or by linker proteins such as ZipA or FtsA, because previous reports have shown multiple divisome protein's genes lacking *E. coli* mutants (not a single gene) only show the severe phenotype of cell division (e.g., Gueiros-Filho, F.J. and Losick R., *Gene Dev*, 16, 2544 (2002)).

Upon inspection of the cryo-EM maps and models, I am not fully convinced by the placement of GMPCPP in two conformations. Superposition of the crystal structure of FtsZ from *S. aureus* in complex with GDP, BeF₃⁻ and Mg²⁺ (Ruiz et al., *PLoS Biol* 2022) onto the current filament shows that Mg²⁺ (and its water coordination sphere) nicely fits the density that is attributed to the GMPCPP gamma-phosphate in conformation 2. Have the authors considered the possibility of modelling only conformation 1 for GMPCPP and adding Mg²⁺ in the density currently attributed to the gamma-phosphate in conformation 2? Besides, Fig. 3b should include a superposition with 7OHK (SaFtsZ with GDP, BeF₃⁻ and Mg²⁺) rather than 3WGN (SaFtsZ with GTPgammaS), as the former is a better mimetic of GTP function.

Re2-6: We appreciate the reviewer's careful inspection. We superimposed our FtsZ on *S. aureus* FtsZ complexed with GDP, BeF₃⁻, and Mg²⁺, and we found that the γ -phosphate part of conformation2 is a bit weak in density, and the Mg²⁺ structure mentioned by the reviewer also overlaps. As the reviewer pointed out, we updated the coordinates with only conformation1 of GMPCPP and Fig. 3, and the new coordinates were registered in PDB. We rewrote in lines 197-202 in the Results section.

Regarding HS-AFM experiments, the authors should clarify how they distinguish single from double filaments in the crowded images shown in Fig. 5. They likely rely on height measurements to differentiate between single and double filaments, but height quantitative data are missing. To strengthen this claim, the authors should provide quantitative height profile analyses and statistical analysis of filament widths and heights. Importantly, experiments shown in Fig. 5 were performed in the presence of GMPPNP (lines 398-399), which the authors' laboratory has showed to form tubes instead of protofilaments (see Fig. 1 in Fujita et al., *Nat Comm* 2023). This raises the question of

248 whether the filaments shown in current Fig. 5 are protofilaments in the straight configuration, as
claimed by the authors (line 401), or tubes. In the latter case, the description and conclusions should
be revisited

Re2-7: To respond to the reviewer's comments, we prepared the height histogram of FtsZ
protofilaments in Extended Data Fig. 12. Our additional experiments yielded the same results with
both GTP and GMPCPP, and based on the height, the filaments on mica were clearly single
protofilaments. As the reviewer pointed out, previous negative-stain images with GMPPNP
showed predominantly helical structures with a height of approximately 200 Å (Fig. 1e in Fujita
et al., Nat Commun 2023). HS-AFM, different from negative staining, observes molecules on the
mica substrate. Our data indicate the absence of a helical structure in HS-AFM.

Reviewer #3 (Remarks to the Author):

The authors employed cryogenic electro-microscopy (cryo-EM) and high-speed atomic force
microscopy (HS-AFM) to investigate the structure and dynamics of FtsZ-ZapA complex. For the cryo-
EM study, the authors resolved an asymmetric ladder-like structure, where ZapA tetramers tethered a
double antiparallel FtsZ protofilament on one side and a single protofilament on the other. Furthermore,
the authors elaborated on detailed structural features underlying interactions between the double
protofilament and the interactions between FtsZ and ZapA molecules, as well as calculated the free
energy to support their conclusions. Using HS-AFM, the authors explored the single-molecule
dynamics of the ZapA-FtsZ complex, employing two approaches: first, by pre-mixing ZapA, FtsZ, and
GTP in a tube prior to HS-AFM imaging, and second, by pre-absorbing FtsZ protofilaments onto a
mica substrate and subsequently adding ZapA to the solution for HS-AFM imaging.

I was particularly invited to technically assess the HS-AFM data and conclusions drawn. While the
HS-AFM data is of high quality, with clearly resolved single molecules, I find the interpretation of
these results less convincing, and believe that further experiments, data analysis, or clarifications are
necessary.

Thank you for your suggestion.

To me, the cryo-EM data clearly indicated an asymmetric ladder-like structure composed of a double
protofilament on the one side and a single protofilament on the other side. However, like the authors
concluded which I agree, the HS-AFM study and cross-sectional height analysis in Extended Data
Fig.7, exclusively show ladder-like structures where ZapA bridges two single FtsZ protofilaments. I
think this discrepancy is indeed surprising and intriguing, but I am not fully convinced that this
observation is an artifact as the authors concluded. Here are my concerns:

Re3-1 We appreciate the reviewer's comments. As this reviewer pointed out, the additional
experiments described below led us to conclude that it is not an artifact.

We re-analyzed the same cryo-EM data of the ZapA-FtsZ complex focusing on ZapA, and

also performed a new cryo-EM analysis of the ZapA dimer mutant (I83E)-FtsZ complex. We
added the results of these analyses in lines 163-194. The analyses revealed multiple distinct
interaction patterns between ZapA and single FtsZ protofilaments, suggesting that the ZapA-FtsZ
interaction is highly dynamic and fluctuating. We also have to consider the constraints of HS-
AFM observations, where molecules are imaged on a two-dimensional mica substrate. Among
those interaction patterns, only molecules compatible with binding on the mica surface must be
observed in HS-AFM, where one is the single protofilament-ZapA-single protofilament
configuration (Fig. 5a-c), and the other is ZapA molecules on double or single protofilaments (Fig.
5f-h). Therefore, we deleted the "artifact" and rewrote the discussion in lines 270-275. Our
additional HS-AFM experiments showed no difference in GTP, GMPCPP, and GMPPNP results.
We accordingly moved Extended Data Fig. 7 to Fig. 5a-c.

1. Did the authors attempt to dilute their pre-mixture samples before mica absorption? It seems that
Extended Data Fig.7 suggests a dense molecular distribution on the mica, whereas the ZapA-FtsZ
complex likely requires more space for proper absorption.

Re3-2: Thank you for your suggestion on additional experiments. We diluted the pre-mixture
sample and observed it on a mica substrate. When the samples were diluted up to 100x, we
observed ladder-like ZapA-FtsZ structures in areas with available space at the 20x dilution. On
the other hand, in the 50x diluted samples ladder-like ZapA-FtsZ structures were hardly observed,
and upon further dilution up to 100x, the FtsZ filaments became significantly shorter and their
adsorption onto the substrate was weak, resulting in the disappearance of the ladder-like structures.
We included the results in Extended Data Fig. 13.

2. As far as I observed, many filaments in Extended Data Fig. 7 appear similar to those FtsZ
protofilaments in Fig 5a-d where, as the authors noted, the absorption surface of FtsZ molecules to the
mica is opposite to the ZapA binding surface. If this is the case, and ZapA, FtsZ, and GTP were pre-
mixed and given sufficient time to form complexes before HS-AFM sample application, why didn't
the authors observe bright dots (ZapA bound to FtsZ) in Extended Data Fig. 7?

Re3-3: We realize that the absence of dots (ZapA) in Extended Data Fig. 7 is likely due to the low
concentration of ZapA in the observation solution and the limited observation area. To avoid
misleading the reader, we replaced Extended Data Fig. 7 with an image where the ZapA
concentration is maintained high and the observation area is broader. This update has been
included in Fig. 5a-c, and lines 217-227.

3. I am not convinced that the distortion upon absorption onto the planar substrates disrupted ZapA-
mediated cross-linking between double protofilaments and single protofilaments and converted them
to cross-linking of single protofilaments. The ladder-like structures in the HS-AFM experiments
(Extended Data Fig. 7) appear too neat, orderly, and clean to result from such disruptions. If the authors'

interpretation is correct, I would expect to observe short, irregular or even broken structures of tethered
single protofilaments

Re3-4: Despite various additional experiments, we did not observe partially broken ladder
structures. Our additional experiments indicate multiple distinct interaction patterns between
ZapA and single FtsZ protofilaments, suggesting that the ZapA-FtsZ interaction is highly dynamic
and fluctuating. We must consider observations' constraints in HS-AFM, where molecules are
imaged on a two-dimensional mica substrate, leading us to conclude that Extended Data Fig. 7
shows the single protofilament-ZapA-single protofilament configuration. We stated this in lines
270-275.

4. The dwell time analysis in Fig.5, indicates that ZapA interacts more strongly with the FtsZ double
protofilament than the single protofilament. Despite the authors's argument that the inter-protofilament
interactions with the double FtsZ filaments is weak, I still find it unlikely that a conversion from double
to single protofilaments, which requires breaking stronger ZapA interactions with two FtsZ and
subsequent formation of new weaker interactions with single FtsZ protofilament, would result in the
neat structures observed in Extended Data Fig.7.

Re3-5: This is an important point. Our additional experiments indicate multiple distinct interaction
patterns between ZapA and single FtsZ protofilaments, suggesting that the ZapA-FtsZ interaction
is highly dynamic and fluctuating. Among the above-described interaction patterns, only
molecules compatible with binding on a mica surface should be observed in HS-AFM, where one
is the single protofilament-ZapA-single protofilament configuration (Fig. 5b-c, j), and the other is
ZapA molecules on double or single protofilaments (Fig. 5f-h). We rewrote lines 217-227 and
discussed it in lines 270-275.

5. It might be also helpful if the authors provided more stats on these protofilaments from the native
complex, or at least more HA-AFM images from different regions (and perhaps additional replicates?).

Re3-6: We prepared wide-field HS-AFM movie of the ZapA-FtsZ complex using ZapA addition
method (Supplementary Video 4).

I think the HS-AFM experiments where the authors first applied FtsZ alone and added ZapA to study
the interaction dwell times are well-conceived. The conclusion that the adsorption surface of FtsZ to
the mica substrate is opposite to the ZapA binding surface is technically sound. However, I think the
data analysis needs refinement to support the claim of increased affinity and provide additional
mechanistic insight into the work. This conclusion could be strengthened by further dwell time analysis,
discriminating all binding events to double protofilaments into two categories: those involving isolated
ZapA molecules and those where ZapA molecules are adjacent. It is apparent in Fig. 5b that the binding
events represent a mixture of (at least) these two cases. As a result, the dwell time histogram in panel
Fig. 5e (at least the right one) should contain more than one time constant (at least two exponentials).

It would also be valuable to explore whether this additional affinity primarily arises from binding to
double protofilaments, as this could provide further novel mechanistic insights into the previously
observed binding cooperativity.

Re3-7: We appreciate the reviewer's comment. We separately analyzed isolated and aligned ZapA
and added the results to Fig. 5i. The analysis emphasizes that ZapA binds much strongly to double
protofilaments when adjacent ZapA molecules are present.

In summary, while the HS-AFM data quality is high, the interpretation and analysis require
improvement to meet the publication standards of Nature Communications. In the current version, I
respectfully believe that the HS-AFM observations of increased affinity in Fig.5 do not present
significant novelty complementing the cryo-EM study, and beyond what has already been observed in
previous studies regarding the cooperative binding. However, with further analysis, these experiments
hold substantial potential to provide valuable insights into the structural dynamics of the system.
Besides, the observation of single FtsZ protofilaments tethered by ZapA molecules is exciting. While
it is premature to conclude that this is an artifact, it certainly warrants further investigation and could
introduce novel insights into the study.

Re3-8: We appreciate the reviewer's suggestion. We have added the ladder growth process to Fig.
5j. The sequential images capture the formation of a short ladder, followed by the extension of one
FtsZ filament, ZapA binding, and the subsequent extension of the other filament. This has been
included in Fig. 5j and Supplementary Video 6, and we wrote lines 245-252.